# The evolution of the vestibular apparatus in apes and humans

**Alessandro Urciuoli[1]\*, Clément Zanolli[2], Amélie Beaudet[3,4], Jean Dumoncel[5], Frédéric Santos[2], Salvador Moyà-Solà[1,6,7], David M Alba[1]\***

[1]Institut Català de Paleontologia Miquel Crusafont, Universitat Autònoma de Barcelona, Cerdanyola del Vallès, Barcelona, Spain; [2]Laboratoire PACEA, UMR 5199 CNRS, Université de Bordeaux, Pessac, France; [3]School of Geography, Archaeology and Environmental Studies, University of the Witwatersrand, Johannesburg, South Africa; [4]Department of Anatomy, University of Pretoria, Pretoria, South Africa; [5]Laboratoire AMIS, UMR 5288 CNRS, Université de Toulouse, Toulouse, France; [6]Institució Catalana de Recerca i Estudis Avançats (ICREA), Barcelona, Spain; [7]Unitat d'Antropologia (Departament de Biologia Animal, Biologia Vegetal i Ecologia), Universitat Autònoma de Barcelona, Cerdanyola del Vallès, Barcelona, Spain

**Abstract** Phylogenetic relationships among extinct hominoids (apes and humans) are controversial due to pervasive homoplasy and the incompleteness of the fossil record. The bony labyrinth might contribute to this debate, as it displays strong phylogenetic signal among other mammals. However, the potential of the vestibular apparatus for phylogenetic reconstruction among fossil apes remains understudied. Here we test and quantify the phylogenetic signal embedded in the vestibular morphology of extant anthropoids (monkeys, apes and humans) and two extinct apes (*Oreopithecus* and *Australopithecus*) as captured by a deformation-based 3D geometric morphometric analysis. We also reconstruct the ancestral morphology of various hominoid clades based on phylogenetically-informed maximum likelihood methods. Besides revealing strong phylogenetic signal in the vestibule and enabling the proposal of potential synapomorphies for various hominoid clades, our results confirm the relevance of vestibular morphology for addressing the controversial phylogenetic relationships of fossil apes.

**\*For correspondence:**
alessandro.urciuoli@icp.cat (AU);
david.alba@icp.cat (DMA)

**Competing interests:** The authors declare that no competing interests exist.

## Introduction

Catarrhine primates (Old World anthropoids) include two extant subclades: cercopithecoids (Old World monkeys) and hominoids (apes and humans). Based on molecular (e.g., *Springer et al., 2012*) and paleontological (e.g., *Harrison, 2013*; *Stevens et al., 2013*) data, both groups diverged during the late Oligocene (≥25 Ma), but experienced very different evolutionary histories. Hominoids first radiated in the early Miocene of Africa (*Harrison, 2010*; *Begun, 2013*; *Begun, 2015*) and subsequently diversified into Eurasia during the middle and late Miocene (*Alba, 2012*; *Begun, 2015*). Their diversity and geographic distribution (humans excluded) was much greater during the Miocene than at present, being currently restricted to a few genera in southeastern Asia and Africa. In contrast, extant cercopithecoid lineages started to diversify later and experienced a major radiation during the late Miocene (*Jablonski and Frost, 2010*), being currently much more diverse and widely distributed than hominoids in both Africa and Asia.

The decimated current diversity of hominoids, coupled with the fragmentary nature of their fossil record, abundant homoplasy (e.g., *Larson, 1998*), and the lack of known fossil hylobatids prior to the latest Miocene (*Harrison, 2016*) make it difficult to confidently infer the phylogenetic relationships of extinct hominoids and thus reliably infer the morphotype of the last common ancestor (LCA)

**eLife digest** Humans, gorillas, chimpanzees, orangutans and gibbons all belong to a group known as the hominoids. This 'superfamily' also includes the immediate ancestors and close relatives of these species, however in many instances the evolutionary relationships between these extinct ape species remain controversial.

While DNA can help evolutionary biologists to work out how living species are related to one another, fossils are typically the principle source of information for extinct species. Inferring evolutionary relationships from fossils must be done with caution, but the bony cavity that houses the inner ear – which is involved in balance and hearing and fairly common in the fossil record – has proven useful for tracing the evolution of certain groups of mammals. However, no one had previously looked to see if this structure could give insights into the evolutionary relatedness among living and extinct hominoids.

Urciuoli et al. have now used a 3D imaging technique to capture the complex shapes of the inner ear cavities of 27 species of monkeys and apes, including humans and two extinct apes (*Oreopithecus* and *Australopithecus*). The results confirmed that the shape of these structures most closely reflected the evolutionary relationships between the species and not, for example, how the animals moved.

Urciuoli et al. went on to identify features of these bony chambers that were shared within several hominoid groups, and to estimate what the inner ears of the ancestors of these groups might have looked like. The results for *Australopithecus*, for example, were consistent with it being most closely related to modern humans than other apes, while those for the enigmatic *Oreopithecus* supported the view that it was a much older species of ape that converged in some respects with other apes still alive today.

The findings highlight the potential of the inner ear for reconstructing the early branches of our family tree. They also offer the prospect of refining the controversial evolutionary relationships within the impressive diversity of extinct ape species.

of various hominoid subclades. This is required not only to adequately understand the evolutionary history of the group as a whole, but also to reconstruct the LCA of chimpanzees and humans, from which the earliest hominins evolved during the late Miocene. The contribution of Miocene apes to debates about hominoid evolution is thus diminished by the numerous controversies about the phylogenetic position of the former.

For example, putative stem hominoids from the early Miocene of Africa (*Begun, 2013*; *Begun, 2015*), such as the proconsulid *Ekembo*, lack most of the synapomorphies of crown hominoids, such that some authors still contend that they might represent stem catarrhines instead (*Harrison, 2010*; *Harrison, 2013*). Even more uncertain is the position of dendropithecids (e.g., *Micropithecus*) and other small-bodied catarrhines from the early to middle Miocene of Africa, which are generally interpreted as stem catarrhines (*Harrison, 2010*; *Harrison, 2013*) but might include stem hominoids as well (*Alba et al., 2015*; *Begun, 2015*). The same controversy applies to the European middle to late Miocene *Pliobates*, recovered as a stem hominoid more derived than proconsulids (*Alba et al., 2015*) or alternatively as a member of the stem catarrhine pliopithecoid radiation (*Nengo et al., 2017*). Regarding undoubted hominoids, the distinction between stem and crown taxa is by no means less controversial, in part due to the virtual lack of fossil hylobatids since their origin in the early Miocene until the latest Miocene (*Harrison, 2016*). This is best exemplified by the late Miocene *Oreopithecus* from Italy, variously considered a hominid (*Moyà Solà and Köhler, 1997*; *Harrison and Rook, 1997*) or a stem hominoid (*Nengo et al., 2017*). Finally, the phylogenetic placement of Miocene apes from Eurasia is also controversial. For example, most Asian forms have been classically considered pongines (*Begun, 2013*), but most recently the late Miocene *Lufengpithecus* from China has been reinterpreted as a stem hominid (*Kelley and Gao, 2012*; *Begun, 2015*). Even more controversial is the position of the middle to late Miocene European dryopithecines, interpreted as either stem hominids (*Alba, 2012*; *Alba et al., 2015*) or hominines (*Begun, 2013*; *Begun, 2015*), and further controversies apply when trying to decipher the phylogenetic relationships among various members of this extinct group.

Deciding among phylogenetic hypotheses for Miocene apes has consequences for our current understanding of hominin origins, from calibrating molecular data to estimate their divergence time to the reconstruction of the ancestral locomotor repertoire from which the earliest bipeds arose. For example, the recently described dryopithecine *Danuvius* from Germany has been used to argue that bipedal and suspensory adaptations characterized the last common ancestor of crown hominids (*Böhme et al., 2019*). However, without a phylogenetic analysis supporting a more basal branching of *Danuvius* compared with the older dryopithecine *Pierolapithecus* (*Moyà-Solà et al., 2004*), which was orthograde but lacked adaptations to both bipedalism and suspension (*Alba et al., 2010*; *Alba, 2012*), the implications for the ancestral condition of the group remain moot.

The recent recovery of enamel proteome sequences from the early Pleistocene (1.9 Ma) fossil pongine *Gigantopithecus* (*Welker et al., 2019*) offers some hope that molecular data will become available for Miocene apes sometime in the future. In the meantime, to better resolve the phylogeny of Miocene hominoids, the search for morphological features not very prone to homoplasy is crucial. Anatomical structures that develop early during development and are not remodeled thereafter (such as the enamel-dentine junction of teeth) represent the best candidates (*Corruccini, 1987*; *Skinner et al., 2008*). In this regard, the inner ear is also a very promising anatomical area (*Spoor and Zonneveld, 1998*; *Spoor et al., 2003*), even if thus far its phylogenetic implications have been mostly explored for fossil hominins only (*Quam et al., 2016*; *Conde-Valverde et al., 2018*; *Ponce de León et al., 2018*; *Beaudet, 2019a*; *Beaudet et al., 2019b*), while its application to fossil apes has been mainly devoted to locomotor inferences (*David et al., 2010*; *Malinzak et al., 2012*; *Rook et al., 2004*; *Ryan et al., 2012*).

Housed in the highly mineralized petrosal bone, which is frequently preserved in the fossil record, the inner ear is composed of a series of endolymph-filled membranes (encased in the corresponding bony labyrinths), namely the cochlea or cochlear duct (involved in hearing) and the vestibular apparatus (devoted to balance and vision stability). The vestibule consists of three (anterior, posterior, and lateral) semicircular canals (SCs) and two macular organs (utricle and saccule). The approximately orthogonal SCs sense angular accelerations and decelerations of the head, while the maculae perceive linear accelerations and thus provide gravity reference (*Spoor and Zonneveld, 1998*; *Rabbitt et al., 2004*; *Johnson Chacko et al., 2018*; *Cheung and Ercoline, 2018*).

Differences in the relative size and morphology of the SCs have been correlated with locomotor agility (*Spoor et al., 1994*; *Spoor et al., 2007*; *Walker et al., 2008*; *Silcox et al., 2009*; *Ryan et al., 2012*; *Perier et al., 2016*) and positional behavior (*Le Maître et al., 2017*), albeit not without criticism (*Rae et al., 2016*; *Benson et al., 2017*; *Coutier et al., 2017*). On the other hand, the bony labyrinth morphology has been considered of great importance for phylogenetic reconstruction in various mammals including hominins (*Quam et al., 2016*; *Conde-Valverde et al., 2018*; *Ponce de León et al., 2018*; *Beaudet, 2019a*; *Beaudet et al., 2019b*) and nonhuman primates (*Lebrun et al., 2010*; *Lebrun et al., 2012*; *Grohé et al., 2016*; *Mennecart et al., 2017*; *Costeur et al., 2018*; *Schwab et al., 2019*). While previous research in hominoids has yielded encouraging results for phylogenetic reconstruction (*Spoor and Zonneveld, 1998*; *Spoor et al., 2003*; *Rook et al., 2004*; *Gunz et al., 2012*), according to some authors phylogeny may not be a major component of ape vestibular morphology (*Le Maître et al., 2017*).

Determining if and to what extent inner ear anatomy reflects phylogeny among extant hominoids is central for assessing the potential of this anatomical area to more confidently resolve the controversial phylogenetic relationships of fossil apes. To provide insight into this question, we use phylogenetically-informed statistical analyses to test the significance and quantify the amount of phylogenetic signal of vestibular shape captured by three-dimensional geometric morphometrics (3DGM) in living hominoids and a broader sample of extant anthropoids. To capture vestibular shape, we mostly rely on a landmark-free, deformation-based 3DGM approach that takes the whole surface into account (*Glaunés and Joshi, 2006*; *Durrleman et al., 2012b*; *Durrleman et al., 2012a*) and enables integrating the spatial trajectory of the semicircular canals with their thickness and volumetric variations, the latter two being more difficult to assess based on mainstream landmark-based 3DGM. Since our results reveal the presence of strong phylogenetic signal in the vestibular morphology, we also employ maximum likelihood methods (*Felsenstein, 1988*; *Schluter et al., 1997*) to reconstruct the ancestral vestibular morphology for the LCA of main hominoid subclades (crown hominoids, hominids, and hominines), with the aim to identify phylogenetically informative characters that can be used in formal cladistic analysis. To test the reliability and illustrate the usefulness of

our approach from a phylogenetic viewpoint, we also include two extinct hominoid taxa: the early hominin *Australopithecus* and the aforementioned controversial late Miocene ape *Oreopithecus*. The well-known phylogenetic placement of *Australopithecus* as the sister-taxon of humans predicts that its vestibular morphology will be somewhat derived towards the modern human condition. On the other hand, our analysis will enable testing the competing phylogenetic hypotheses for *Oreopithecus* (stem hominoid vs. hominid), thereby illustrating the potential of our method for clarifying the controversial affinities of extinct apes.

## Results

### Semicircular canal shape variation among anthropoids

In spite of a similar spatial configuration of the three SCs, their trajectory, stoutness and relative proportions are quite variable among anthropoids (*Figure 1*). A bgPCA performed among major anthropoid groups (platyrrhines, cercopithecoids, hylobatids, and hominids) enables their accurate distinction (*Figure 2*, *Figure 2—figure supplement 1a*, *Figure 2—source data 1*), as shown by classification results (more than 95% individuals correctly classified after cross-validation; *Table 1*). In particular, bgPC1 discriminates hylobatids from hominids and all the remaining taxa (*Figure 2*, *Figure 2—figure supplement 1c*). A landmark-based analysis applied to the same sample yields very similar results except for hylobatids, due to the reasons explained in the next section. Shape variation in the analyzed sample accounts for a strong phylogenetic signal ($K_{mult}$ = 1.248, p<0.001), and this also holds for the first three bgPCs separately (see below).

When analyzed individually, we recover a similarly strong phylogenetic signal ($\lambda$ = 1 and $K$ = 1.15; *Table 2*) for bgPC1 (68.8% of total variance). bgPC1 captures differences in thickness and cross-section of the SCs, and is also driven by the development of the macular organs relative to the canals. Great apes fall on positive values (*Figure 2*, *Figure 2—figure supplement 1a*) due to their stout and flattened SCs combined with an extensive vertical compression of the anterior canal, a more anterosuperior insertion of the lateral canal into the vestibule, and a greater volume of the vestibular recesses relative to the canals (*Figures 3a–e*, *4a–e* and *5a–e*). Hylobatids, as well as colobine and papionin cercopithecoids, showing slender and elongated canals but maintaining well developed ampullae, largely overlap on negative values, while cercopithecins and platyrrhines display intermediate values due to their slightly inflated SCs (*Figures 2*, *3f–h*, *4f–h* and *5f–h*). bgPC2 (19.6% of total variance) also bears strong phylogenetic signal ($\lambda$ = 0.91; $K$ = 1.51; *Table 2*), with variance accumulating among rather than within clades (as indicated by $K$ > 1). This axis separates platyrrhines—especially *Ateles* (*Figure 1i*)—from other anthropoids due to the more reduced lateral canal in the former, which is inversely proportional to anterior canal development and vertical elongation (*Figure 2a*). In contrast, *Gorilla* (*Figures 1a*, *3a*, *4a* and *5a*) occupies the opposite end of the distribution due to its large lateral canal and reduced anterior one, the latter being also vertically compressed, whereas the remaining hominoids show intermediate values along bgPC2. bgPC3 (11.6% of variance), which is driven by both trajectory and relative size of the SCs (*Figure 2b*), still displays a strong and significant phylogenetic signal ($\lambda$ = 1 and $K$ = 1.16; *Table 2*). Hylobatids (*Figures 1f–h*, *3f–h*, *4f–h* and *5f–h*) display the highest scores for bgPC3, as a result of the right to acute angle formed by the apex of the common crus (CC), the latter being also shorter, a more anterosuperiorly projecting and long anterior canal, an obtuse angle between the planes identified by the anterior and lateral canals, and a more developed lateral canal relative to the posterior one, which is also posteriorly oriented. Most of the taxa fall within moderate positive and moderate negative values, with some cercopithecoids (*Cercopithecus*, *Macaca*, *Papio* and *Nasalis*, among others) and *Cebus* occupying the negative end of the distribution. African apes, cercopithecines, and *Nasalis* occupy an intermediate position, being characterized by a well-developed and anteriorly inclined posterior canal, an obtuse to right angle of the CC apex, and an obtuse angle between the vertical canals, combined with a larger lateral canal. Orangutans fall on moderately positive values and are distinguished from other great apes and humans by an anterosuperiorly projecting anterior canal (less than in hylobatids).

When fossil specimens are plotted a posteriori onto the tangent space identified by extant taxa, *Oreopithecus* (BAC 208; *Figures 1n* and *6a*) falls on moderately positive scores for bgPC1 (where the distributions of hominids, cercopithecins, and platyrrhines overlap), while the two

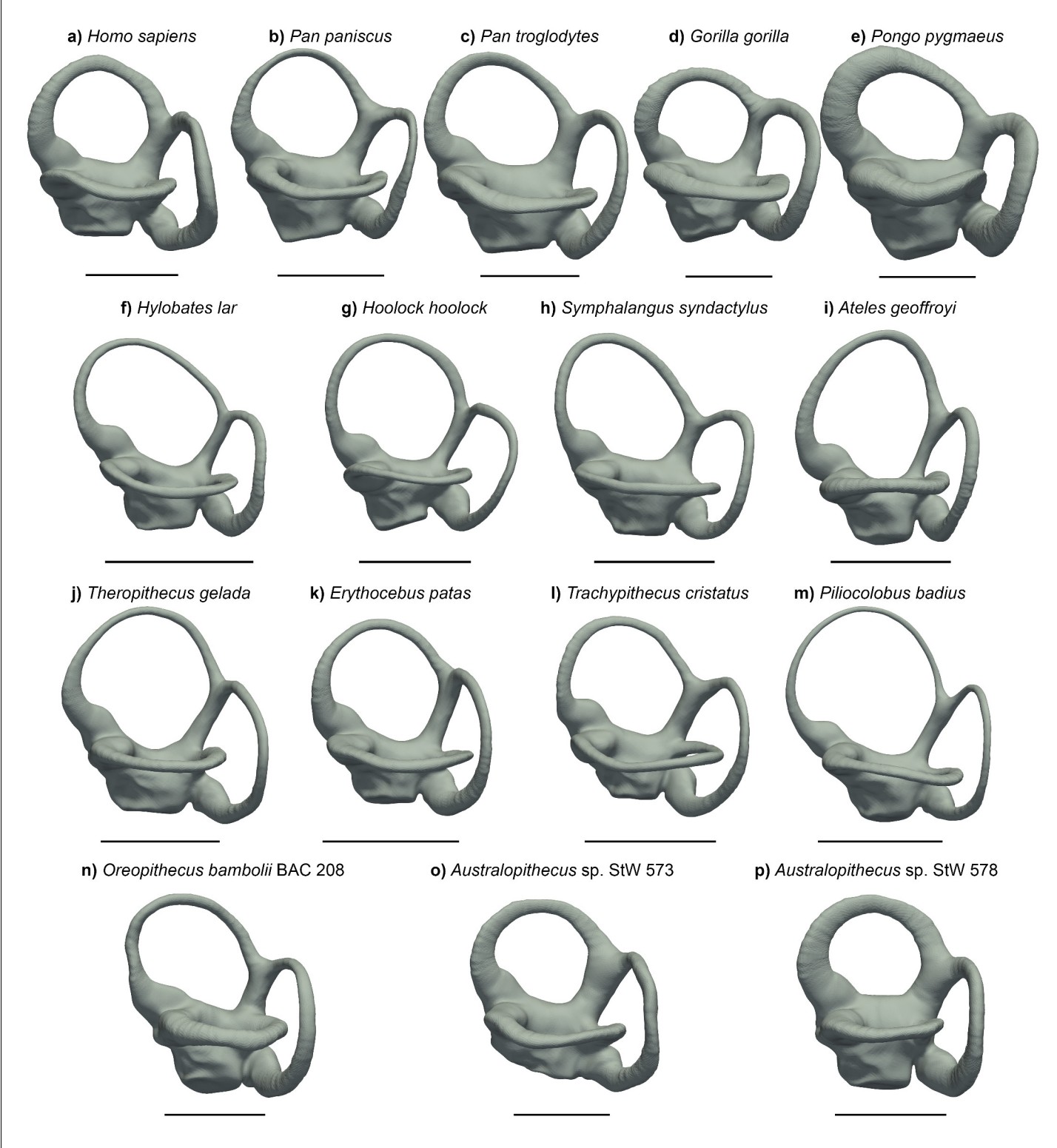

**Figure 1.** The vestibular apparatus of extant hominoids, other anthropoids, and some fossil hominoids, in lateral view. (**a**) *Homo sapiens* (EMBR 121); (**b**) *Pan paniscus* (MCZ 38019); (**c**) *Pan troglodytes* (AMNH.M 51204); (**d**) *Gorilla gorilla* (AMNH.M 167338); (**e**) *Pongo pygmaeus* (IPS 10647); (**f**) *Hylobates lar* (MCZ 41424); (**g**) *Hoolock hoolock* (AMNH.M 83425); (**h**) *Symphalangus syndactylus* (AMNH.M 106583); (**i**) *Ateles geoffroyi* (MCZ 29628); (**j**) *Theropithecus gelada* (AMNH.M 60568); (**k**) *Erythrocebus patas* (MCZ 47017); (**l**) *Trachypithecus cristatus* (MCZ 35597); (**m**) *Piliocolobus badius* (MCZ 24793); (**n**) *Oreopithecus bambolii* (BAC 208); (**o**) *Australopithecus* sp. (StW 573); (**p**) *Australopithecus* sp. (StW 578). Scale bars equal 5 mm.

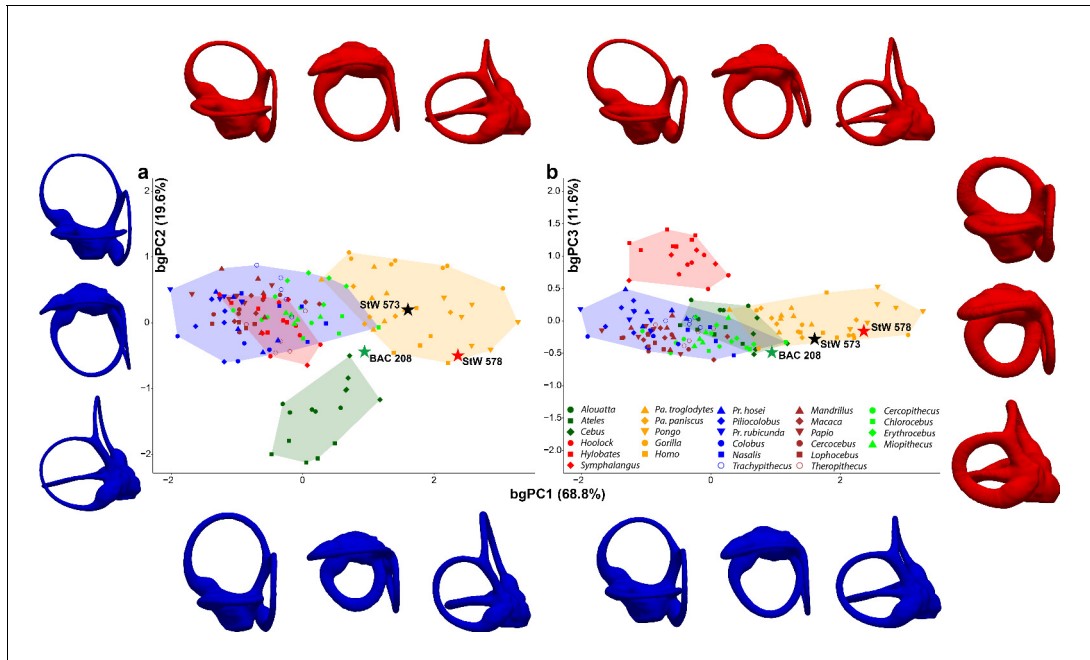

**Figure 2.** Main patterns of vestibular shape variation among the analyzed anthropoid sample as shown by bivariate plots of principal components from the between-group principal components analysis (bgPCA) using a few major clades (i.e., platyrrhines, cercopithecoids, hylobatids and hominids) as grouping factor. (**a**) bgPC2 vs. bgPC1. (**b**) bgPC3 vs. bgPC1. Variance explained by each bgPC is included within parentheses. Color code: dark green, platyrrhines; orange, hominids; red, hylobatids; brown, papionins; green, cercopithecins; blue, colobines. Colored stars correspond to: green, *Oreopithecus*; black, *Australopithecus* sp. (StW 573); red *Australopithecus* sp. (StW 578). Lateral (top/left), superior (middle), and posterior (bottom/right) views of deformation maximum (red) and minimum (blue) conformations for each bgPC are shown along each axis. Hominids are distinguished from hylobatids and cercopithecoids along bgPC1 (positive vs. mostly negative values, respectively), which is mainly driven by the volumetric proportions of the SCs and by their size relative to that of the vestibular recesses. bgPC2, driven by the size of the anterior and posterior SCs relative to the lateral one, distinguishes platyrrhines (more negative values) from catarrhines. Hylobatids (on positive values) differ from all other anthropoids along bgPC3 due to the reduced and posteriorly tilted posterior SC, as well as by the relative orientation among the canals. *Oreopithecus* does not match the variability of extant anthropoids as its morphology shows a mosaic of primitive and derived features. The two *Australopithecus* specimens match instead the range of extant hominids, with StW 573 being most similar to *Pan*, while StW 578 to *Homo*.

The online version of this article includes the following source data and figure supplement(s) for figure 2:

**Source data 1.** Individual scores for all the principal components (bgPC) yielded by the between-group principal components analysis (bgPCA) of deformation-based 3DGM of vestibular shape for anthropoids, using major taxa (i.e., hominids, hylobatids, cercopithecoids, and platyrrhines) as grouping factor.

**Figure supplement 1.** Box-and-whisker plots of the principal components (bgPCs) from the between-group principal components analyses (bgPCA) of vestibular shape for the anthropoid sample.

*Australopithecus* individuals (StW 573 and StW 578; **Figures 1o,p** and **6b–c**) fall within the range of living great apes and humans (**Figure 2**, **Figure 2—figure supplement 1a**). This is due to the volumetric proportions of their SCs and the possession of voluminous vestibular recesses (although the latter character is less pronounced in *Oreopithecus*). In BAC 208, SC volume is greater on the lateral

**Table 1.** Percentage of correctly classified individuals with cross-validation according to the groups (hominids, hylobatids, cercopithecoids, platyrrhines) used in the between-group principal components analysis based on group-centroid distances.

|  | Cercopithecoidea | Hominidae | Hylobatidae | Platyrrhini |
|---|---|---|---|---|
| Cercopithecoidea | 96.3% | 3.8% | 0.0% | 0.0% |
| Hominidae | 3.3% | 96.7% | 0.0% | 0.0% |
| Hylobatidae | 5.9% | 0.0% | 94.1% | 0.0% |
| Platyrrhini | 0.0% | 0.0% | 0.0% | 100.0% |

**Table 2.** Phylogenetic signal results for a between-group principal components analysis (bgPCA) applied to vestibular shape deformation fields in the analyzed sample of extant anthropoids.

|  | bgPC1 | bgPC2 | bgPC3 |
|---|---|---|---|
| Variance | 68.79% | 19.60% | 11.61% |
| Eigenvalue | 0.821 | 0.234 | 0.138 |
| Pagel's λ | 1.000 (p<0.001) | 0.921 (p<0.001) | 1.000 (p<0.001) |
| Blomberg's K | 1.152 (p<0.001) | 1.514 (p<0.001) | 1.163 (p<0.001) |

canal, while the two *Australopithecus* specimens possess stouter vertical canals. When the relative size of the SCs is taken into account, StW 573 shows more evenly developed canals than StW 578 and BAC 208, which both display a smaller lateral canal. Their position along the bgPC2 axis reflects these differences, with StW 573 falling on positive scores (close to the mean value for hominids and cercopithecoids; *Figure 2a*, *Figure 2—figure supplement 1b*) and the other two specimens occupying moderately negative values (within the range of extant catarrhines and approaching that of platyrrhines; *Figure 2a*, *Figure 2—figure supplement 1b*). This is caused by the comparatively smaller lateral canal and by the large vertical canal in StW 578 and in BAC 208. Due to the acute angles between the planes identified by the anterior and lateral canals and that between the planes of the posterior and anterior canals, *Oreopithecus* falls at the negative end of the extant anthropoid distribution for the bgPC3 (*Figure 2b*, *Figure 2—figure supplement 1c*). On the other hand, the two *Australopithecus* specimens occupy more intermediate values because of the possession of a right angle between the planes of the aforementioned canals.

When the bgPCs are considered at the same time (*Figure 2*), the australopith specimens fall well within the great ape and human range. This is further supported by their posterior probabilities of

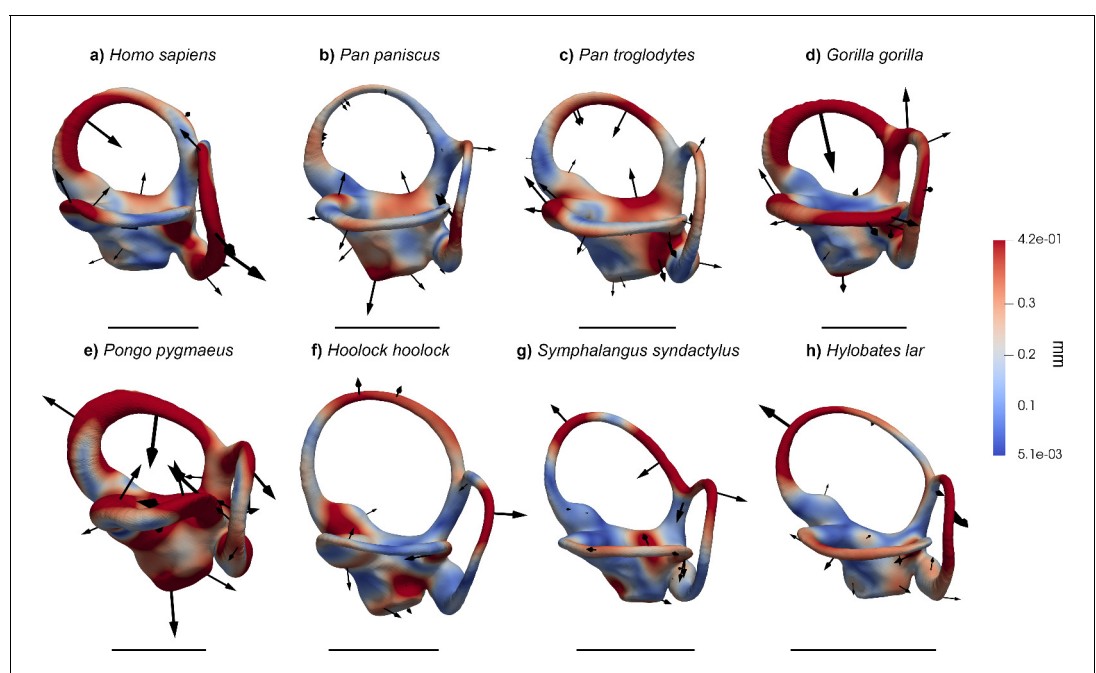

**Figure 3.** Results of the 3DGM deformation-based analysis superimposed on the vestibular apparatus of hominoids in lateral view. Cumulative displacement variations are rendered by pseudocolor scale ranging from dark blue (5.1 µm) to dark red (0.42 mm). Black arrows correspond to the vectors identifying the direction and amount of displacement. (a) *Homo sapiens* (EMBR 121); (b) *Pan paniscus* (MCZ 38019); (c) *Pan troglodytes* (AMNH. M 51204); (d) *Gorilla gorilla* (AMNH.M 167338); (e) *Pongo pygmaeus* (IPS10647); (f) *Hoolock hoolock* (AMNH.M 83425); (g) *Symphalangus syndactylus* (AMNH.M 106583); (h) *Hylobates lar* (MCZ 41424). Scale bars equal 5 mm. Hominids (a–e) differ from other anthropoids in the stouter semicircular canals and the vertically compressed (more eccentric) anterior semicircular canal, while hylobatids (f–h) possess slender canals more similar to those of cercopithecoids (j–m).

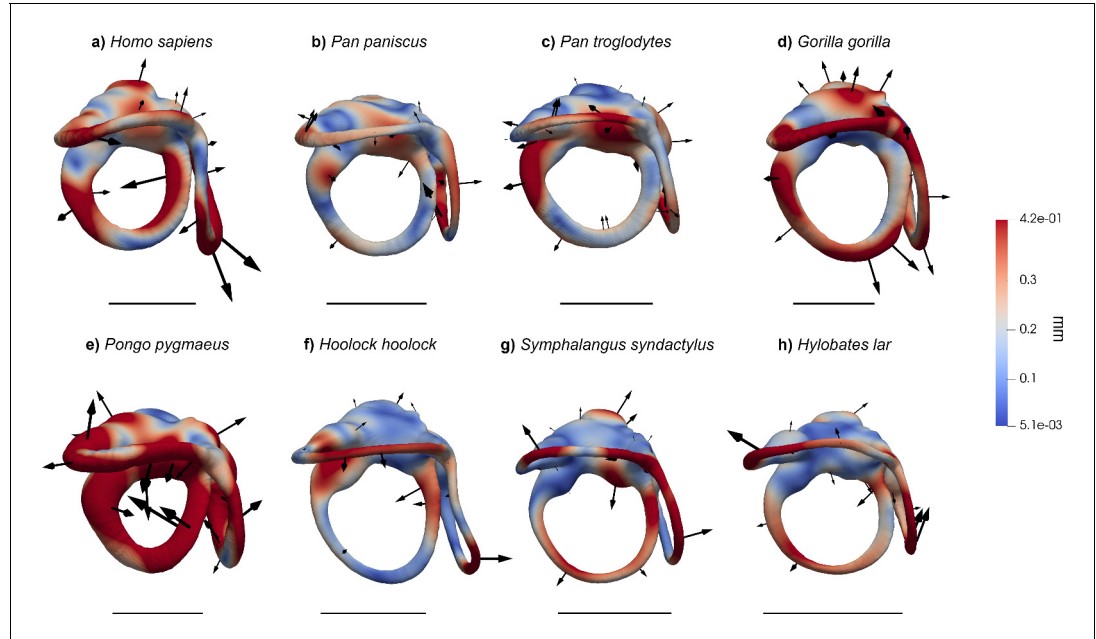

**Figure 4.** Results of the 3DGM deformation-based analysis superimposed on the vestibular apparatus of hominoids, in superior view. Cumulative displacement variations are rendered by pseudocolor scale ranging from dark blue (<5.1 μm) to dark red (0.42 mm). Black arrows correspond to the vectors identifying the direction and amount of displacement. (**a**) *Homo sapiens* (EMBR 121); (**b**) *Pan paniscus* (MCZ 38019); (**c**) *Pan troglodytes* (AMNH. M 51204); (**d**) *Gorilla gorilla* (AMNH.M 167338); (**e**) *Pongo pygmaeus* (IPS10647); (**f**) *Hoolock hoolock* (AMNH.M 83425); (**g**) *Symphalangus syndactylus* (AMNH.M 106583); (**h**) *Hylobates lar* (MCZ 41424). Scale bars equal 5 mm. Hominids (**a–e**) differ from other anthropoids in the stouter semicircular canals, except for the lateral canal of *Ateles*. Among great apes, *Pongo* (**e**) displays the most inflated canals and a uniquely displaced lateral semicircular canal, *Pan* (**b, c**) has the least derived morphology, and *Gorilla* (**e**) possesses a laterally protruding lateral canal. *Homo* (**a**) combines a slight (variable) reduction of the lateral canal with an enlarged posterior canal. Hylobatids (**f–h**) show an obtuse angle between the posterior and the anterior SCs.

group membership based on the proximity of fossil specimen scores to groups centroids, with StW 573 and 578 being classified as hominids with p=0.678 and p=0.190, respectively (*Table 3*). StW 573 falls close to *Pan* and *Homo*, whereas StW 578 occupies an intermediate position between humans and orangutans due to its stouter volumetric proportions. When the posterior probabilities are computed using the centroids of the hominoid genera, StW 573 is classified as *Pan* as first option (p=0.368) and as *Homo* as second (p=0.264), while StW 578 is more clearly classified as *Homo* with p=0.727 (*Table 4*). These results suggest that both australopith specimens show vestibular similarities with extant humans, but that StW 573 display a more plesiomorphic (chimpanzee-like) morphology. In turn, *Oreopithecus* shows a mosaic of vestibular features (*Pan*-like volumetric proportions, small lateral canal, and acute angles between the anterior and both the posterior and the lateral canals) that does not match the condition of any extant taxon (*Figure 2*). The posterior probabilities indicate closest similarities with cercopithecoids, followed by hominids and platyrrhines, although in all instances it falls outside the variability of the extant members of these groups (p<0.05; *Table 3*). When comparisons are restricted to hominoid genera, *Oreopithecus* appears more similar to humans than to any ape genus, but again with a posterior probability that indicates significant differences with all of them (p<0.05; *Table 4*).

The multivariate regression between shape (deformation fields) and size (log-transformed volume of the vestibule) shows a significant correlation (i.e., allometry) at p<0.001, but nevertheless explains only a limited portion of the variance ($R^2$ = 0.192). Bivariate regressions of the bgPCs against log-transformed cube root of vestibular volume reveal a significant correlation only for bgPC1 ($R^2$ = 0.635, p<0.001, *Table 2*). A visual inspection of the scatter of points (*Figure 7a*) suggests that allometry for bgPC1 is more marked in hominids. This is confirmed when separate regressions are performed for hominids ($R^2$ = 0.480, p<0.001) and the rest of the sample ($R^2$ = 0.058, p<0.01), with the former displaying also a higher slope (*Table 5*). When phylogeny is considered by means of PGLS regression (*Table 5*), the correlation for the whole sample is still significant but explains much

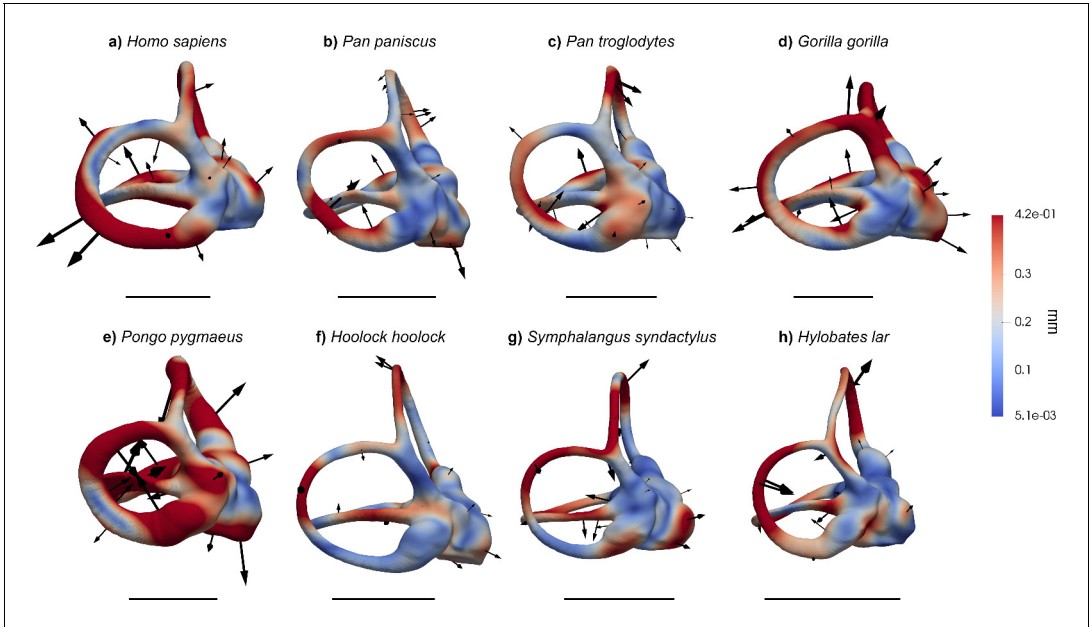

**Figure 5.** Results of the 3DGM deformation-based analysis superimposed on the vestibular apparatus of hominoids in posterior view. Cumulative displacement variations are rendered by pseudocolor scale ranging from dark blue (<5.1 μm) to dark red (0.42 mm). Black arrows correspond to the vectors identifying the direction and amount of displacement. (a) *Homo sapiens* (EMBR 121); (b) *Pan paniscus* (MCZ 38019); (c) *Pan troglodytes* (AMNH. M 51204); (d) *Gorilla gorilla* (AMNH.M 167338); (e) *Pongo pygmaeus* (IPS10647); (f) *Hoolock hoolock* (AMNH.M 83425); (g) *Symphalangus syndactylus* (AMNH.M 106583); (h) *Hylobates lar* (MCZ 41424). Scale bars equal 5 mm. *Homo* (a), together with *Gorilla* (d), displays a markedly posterolaterally protruding posterior canal. The two species of *Pan* (b, c) can be distinguished from one another for the orientation of the anterior canal (more medially inclined in *P. troglodytes*, (c). Hylobatids (f–h) display a small posterior canal relative to the size of the anterior and lateral ones.

less variance ($R^2$ = 0.261, p<0.01), becoming non-significant for hominids and the rest of the sample separately. bgPC3 shows a low yet significant correlation with volume for non-hominids, which becomes non-significant after PGLS correction (*Table 5*). Both *Australopithecus* and *Oreopithecus* overlap with the hominid scatter of points, with the two australopith specimens falling above the hominid regression line, whereas BAC 208 falls slightly below (although well above that of non-hominid anthropoids).

The bivariate regression between the log-transformed cube root of the SC volume and SC length shows in all instances a significant correlation that nevertheless only explains a limited (ca. 20–30%) amount of variance (*Table 5*, *Figure 7b*). Isometry cannot be rejected for anthropoids as a whole, but a negatively allometric relationship emerges (revealing that length increases faster than volume) when hominids and other taxa are analyzed separately (*Table 5*). The latter is confirmed by PGLS regressions for the whole sample and the two groups separately, which further explain a higher proportion of variance (*Table 5*), although the hominid regression is not significant with all probability due to small sample size. The bivariate plot (*Figure 7b*) shows an allometric grade shift between hominids and the remaining anthropoid taxa, which is confirmed by ANCOVA results—indicating no significant differences (F = 0.705, p=0.403) between the allometric slopes but significantly different intercepts (F = 263.26, p<0.001) between the two groups. This indicates that hominids possess more voluminous (i.e., stouter) canals than other anthropoids at equal lengths, with only minimal overlap. All the fossil specimens display hominid-like volumetric proportions (*Figure 7b*): StW 573 and 578 fall slightly above the hominid regression line, whereas BAC 208, due to its slenderer semicircular canals, falls below (although much closer than to the non-hominid regression line).

## Exploration of a preexisting group structure in the tangent space of the vestibular shape

Recently, caution has been advised regarding the use of between-group PCA (bgPCA) applied to 3D geometric morphometric (GM) data, as it could produce spurious grouping when there are fewer groups than variables (*Cardini et al., 2019*). However, the same study also highlighted that the

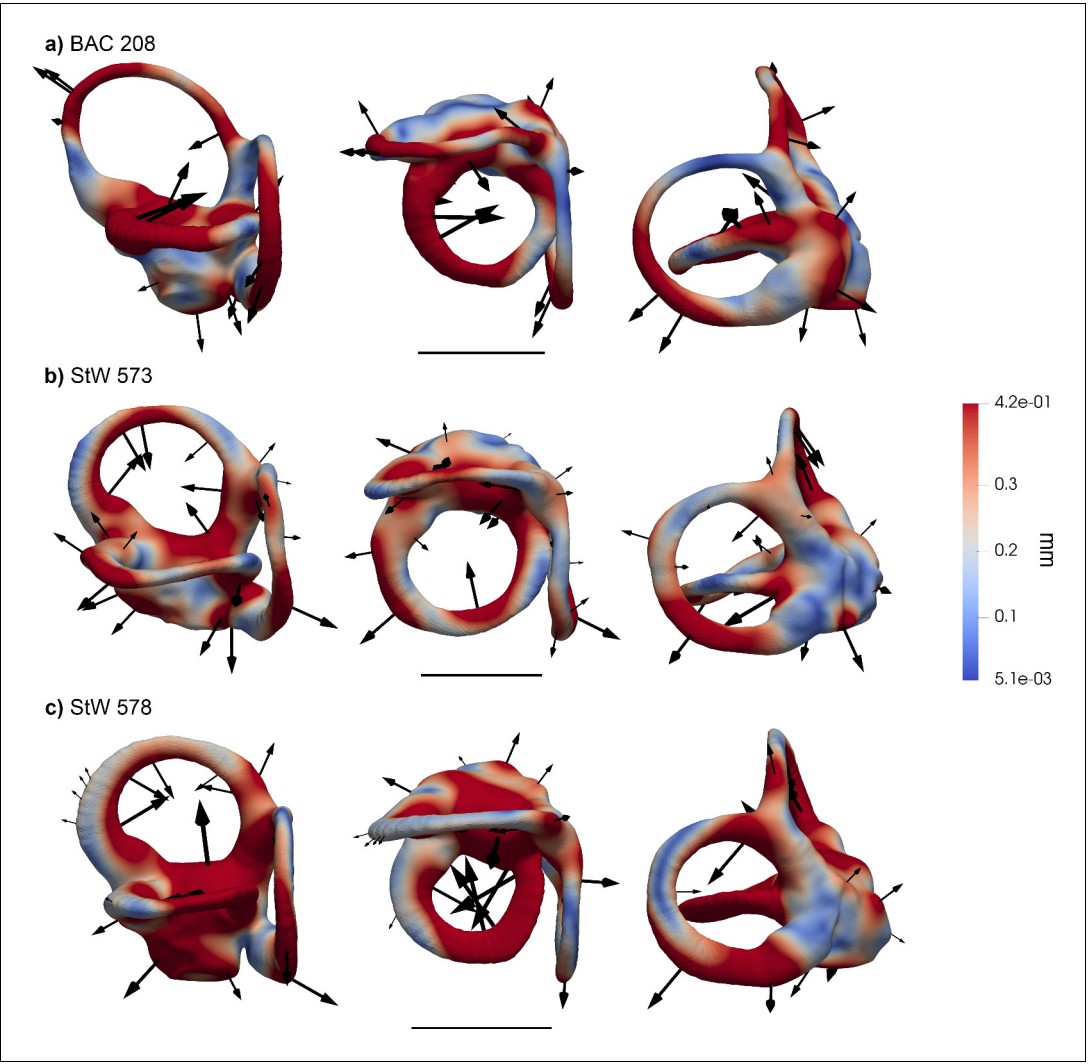

**Figure 6.** Results of the 3DGM deformation-based analysis superimposed on the vestibular apparatus of the fossil specimens included in the present study. Each vestibule is displayed in lateral (left), superior (middle), and posterior (right) views. Cumulative displacement variations are rendered by pseudocolor scale ranging from dark blue (<5.1 μm) to dark red (0.42 mm). Black arrows correspond to the vectors identifying the direction and amount of displacement. (a) *Oreopithecus bambolii* (BAC 208); (b) *Australopithecus* sp. (StW 573); (c) *Australopithecus* sp. (StW 578). Scale bars equal 5 mm. *Oreopithecus* (a) displays large vertical canals and a small, flat lateral one that intersect the plane of the posterior SC. The angle between the anterior and posterior SCs is one of the narrowest among the studied sample. StW 573 (b) and StW 578 (c) differ by means of a larger lateral canal in the previous and by the more rounded and developed anterior SC in the latter.

presence of a strong covariance among the variables (as found in many biological structures) largely reduces the magnitude of the problem. Interestingly, due to the properties of diffeomorphisms, the set of momenta is expected to be highly correlated, as close momenta tend to covary.

Prior to computing the bgPCA, we explored the principal components resulting from the vestibular shape GM analyses to investigate the presence of a preexisting group structure, which was found to be similar to that showed by bgPCA for both extant and fossil taxa (*Figure 8*). We used hierarchical clustering analysis (HCA) on the deformation fields for assessing the probability of correct classification of individuals according to the groups used in the bgPCA. The confusion matrix resulting from the HCA shows that most individuals are correctly identified in the corresponding groups (*Table 6*). Only in the deformation-based analyses hylobatids show a low percentage of classification

**Table 3.** Posterior probabilities of group membership based on the bgPC scores for fossil specimens in the analysis based on the anthropoid sample.
Note that these are probability estimates of having a particular score given membership in a particular group, not the likelihood of group membership in each of a priori defined groups given a particular score. Highest probability for each specimen in bold.

|  | Cercopithecoidea | Hominidae | Hylobatidae | Platyrrhini |
|---|---|---|---|---|
| BAC 208 (*Oreopithecus*) | **p=0.046** | p=0.013 | p<0.001 | p=0.012 |
| StW 573 (*Australopithecus*) | p=0.005 | **p=0.678** | p<0.001 | p<0.001 |
| StW 578 (*Australopithecus*) | p<0.001 | **p=0.190** | p<0.001 | p<0.001 |

(24% of the individuals), mostly due to the great similarity in the volumetric proportions and surface shape of the vestibule between this group and cercopithecoids.

Another way to ascertain that the group separation observed in our bgPCA is not the result of any bias is to compare different kind of analyses with a same data set and the same grouping factor. We thus compared our deformation-based results with those obtained from a configuration of 3D semilandmarks commonly used to investigate the vestibular shape (*Gunz et al., 2012*). Both analyses, based on the full primate sample, yielded a similar group separation (compare *Figure 2* with *Figure 9*), as shown by the components resulting from the bgPCA. These results are also coherent with the biological reality, enabling the discrimination of major anthropoid clades in agreement with their phylogenetic relationships. In the landmark-based approach, hylobatids largely overlap with great apes in both bgPC1 (occupying an intermediate position) and bgPC2 (*Figure 9a*), but can be distinguished from them (and other anthropoids) to a large extent based on bgPC3 (*Figure 9b*). Shape variation in the analyzed sample, as captured by landmark-based 3DGM, accounts for a strong phylogenetic signal ($K_{mult}$ = 0.973, p<0.001), and this also holds for the first three bgPCs separately (see below).

Shape differences along bgPC1 (53.3% of total variance) in the landmark-based approach embed a strong phylogenetic signal ($\lambda$ = 1 and $K$ = 1.26; *Table 7*). This component correlates with the insertion of the lateral canal on the vestibule, the size and shape of the posterior canal, and the roundness of the SCs. Great apes and humans fall on negative values for the first axis (*Figure 9c*), as they are characterized by smaller canals compared to the size of the vestibular recesses and less rounded SCs (particularly the anterior one, which is vertically compressed). Hylobatids stand on intermediate scores that largely overlap with the hominid range (*Figure 9*) due to a combination of long SCs, a vertically compressed anterior canal, and well separated lateral and posterior canals (since posterior canal is posteriorly displaced and the lateral canal inserts anteriorly in the vestibule). In contrast, Old world monkeys tend to be located on positive values of bgPC1 and display a protruding lateral canal that intersects the plane defined by the posterior canal.

bgPC2 (29.3% of variance) separates platyrrhines—especially *Ateles* found in the most negative scores (*Figure 1i*)—from other anthropoids due to the more reduced lateral canal in the former, which is inversely proportional to anterior canal development and vertical elongation (*Figure 9d*). This pattern is shared, although to a lesser extent, by humans (*Figure 1d*) and *Theropithecus* (*Figure 1j*), which possess more developed anterior and posterior canals relative to the lateral one.

**Table 4.** Posterior probabilities of group membership based on the bgPC scores for fossil specimens in the analysis restricted to hominoid genera.
Note that these are probability estimates of having a particular score given membership in a particular group, not the likelihood of group membership in each of a priori defined groups given a particular score. The highest probability for each specimen in bold.

|  | *Hoolock* | *Hylobates* | *Symphalangus* | *Pongo* | *Gorilla* | *Pan* | *Homo* |
|---|---|---|---|---|---|---|---|
| BAC 208 (*Oreopithecus*) | p<0.001 | p<0.001 | p<0.001 | p<0.001 | p<0.001 | p=0.002 | **p=0.026** |
| StW 573 (*Australopithecus*) | p<0.001 | p<0.001 | p<0.001 | p=0.019 | p<0.001 | **p=0.368** | p=0.264 |
| StW 578 (*Australopithecus*) | p<0.001 | p<0.001 | p<0.001 | p=0.062 | p=0.077 | p=0.030 | **p=0.727** |

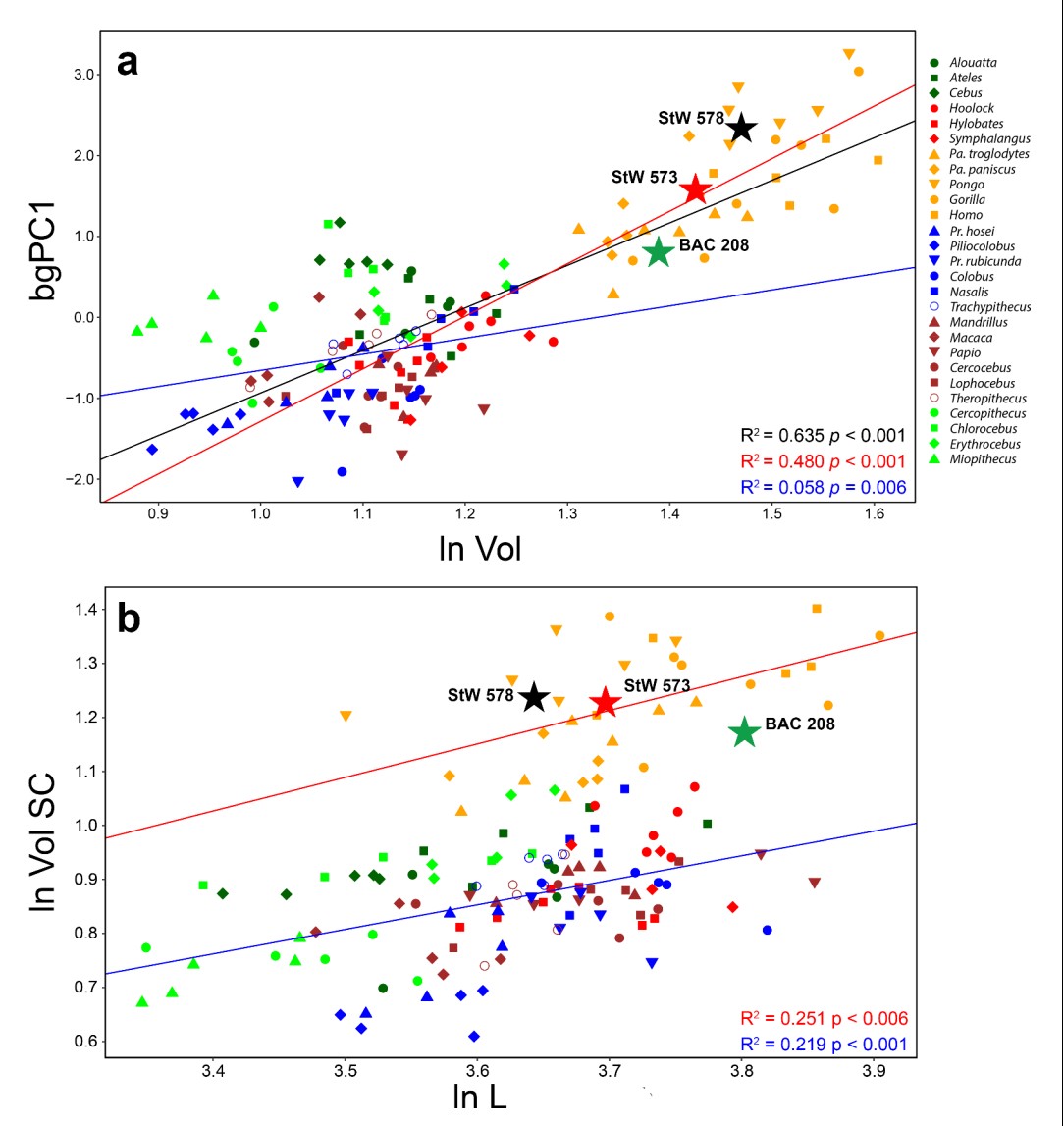

**Figure 7.** Bivariate regressions between (**a**) vestibular shape (as represented by bgPC1) and log-transformed cube root of vestibular volume (ln Vol), and (**b**) semicircular canal log-transformed cube root of volume (ln VolSC) vs.log-transformed length (ln L). Lines represent OLS best-fit lines for the whole anthropoid sample (black), hominids (red), and other anthropoid taxa (blue). Note that there is a significant correlation between bgPC1 (which captures differences in SC thickness and macular organ size) and vestibular volume, more marked in hominids than in the rest of the sample (see *Table 3* for further details). Note as well that hominids and other anthropoids show a similar negatively allometric relationship between the cube root of volume and length of the SCs, but with a marked allometric grade shift—such hominids possess stouter canals than other anthropoid taxa at comparable lengths once size-scaling effects have been taken into account (see *Table 3* for further details). Both australopiths fall above the regression line for hominids, while *Oreopithecus* is slightly below, yet well above the regression line for non-hominid taxa. Color code as in *Figure 2*.

The online version of this article includes the following source data for figure 7:

**Source data 1.** Linear measurements for the analyzed specimens and used for computing the linear regressions.

In contrast, *Gorilla* (*Figure 1a*) occupies the positive end of the distribution due to its large lateral canal and reduced anterior one, the latter being also vertically compressed, whereas the remaining hominoids show intermediate values along bgPC2. The identified phylogenetic signal displayed by

**Table 5.** Bivariate regressions of vestibular shape vs. volume and of semicular canal volume vs. length.

Both ordinary least-square linear regressions (OLS) and phylogenetic generalized least-square regressions (PGLS) are provided for the whole anthropoid sample, as well as hominids and non-hominids separately. Vestibular shape is represented by the first three principal components (bgPC), while vestibular volume (Vol) is represented by its log-transformed cube root. Semicular canal volume (VolSC) and length (L) are represented by the log-transformed cube root and the log-transformed length, respectively, of the three semicircular canals together. For each regression, the coefficient of determination ($R^2$), the significance of the slope (p), and the slope and intercept values with their corresponding standard error (SE) and 95% confidence intervals (CI) are included. Regressions bolded when significant at $p<0.05$. For bgPCs vs. VOL regressions, a significant correlation denotes allometry, while for VOLSC vs. L regressions there is allometry when the correlation is significant and the 95% CI for the slope excludes unity.

| | $R^2$ | p | slope | SE | 95% CI | | intercept | SE | 95% CI | |
|---|---|---|---|---|---|---|---|---|---|---|
| **OLS** | | | | | | | | | | |
| Anthropoids (n = 142) | | | | | | | | | | |
| **bgPC1 vs. ln Vol** | **0.635** | **<0.001** | **5.257** | **0.335** | **4.600** | **5.914** | **−6.191** | **0.399** | **−6.973** | **−5.409** |
| bgPC2 vs. ln Vol | 0.008 | 0.146 | 0.453 | 0.309 | −0.154 | 1.060 | −0.533 | 0.368 | −1.254 | 0.188 |
| bgPC3 vs. ln Vol | 0.000 | 0.385 | 0.192 | 0.220 | −0.239 | 0.622 | −0.226 | 0.261 | −0.738 | 0.287 |
| **ln VolSC vs. ln L** | **0.288** | **<0.001** | **0.897** | **0.118** | **0.666** | **1.128** | **−2.328** | **0.429** | **−3.169** | **−1.487** |
| Hominids (n = 30) | | | | | | | | | | |
| **bgPC1 vs. ln Vol** | **0.480** | **<0.001** | **6.496** | **1.233** | **4.079** | **8.913** | **−7.781** | **1.797** | **−11.303** | **−4.259** |
| bgPC2 vs. ln Vol | 0.026 | 0.195 | −1.400 | 1.054 | −3.466 | 0.666 | 2.307 | 1.538 | −0.708 | 5.322 |
| bgPC3 vs. ln Vol | 0.039 | 0.153 | −0.720 | 0.490 | −1.681 | 0.240 | 1.036 | 0.714 | −0.365 | 2.436 |
| **ln VolSC vs. ln L** | **0.251** | **0.003** | **0.621** | **0.190** | **0.249** | **0.992** | **−1.084** | **0.705** | **−2.465** | **0.297** |
| Non-hominids (n = 112) | | | | | | | | | | |
| **bgPC1 vs. ln Vol** | **0.058** | **0.006** | **1.990** | **0.709** | **0.601** | **3.380** | **−2.645** | **0.785** | **−4.183** | **−1.108** |
| bgPC2 vs. ln Vol | 0.010 | 0.152 | −1.027 | 0.712 | −2.422 | 0.368 | 1.061 | 0.788 | −0.482 | 2.605 |
| **bgPC3 vs. ln Vol** | **0.046** | **0.013** | **1.332** | **0.529** | **0.295** | **2.368** | **−1.466** | **0.585** | **−2.613** | **−0.319** |
| **ln VolSC vs. ln L** | **0.219** | **<0.001** | **0.454** | **0.080** | **0.297** | **0.611** | **−0.783** | **0.291** | **−1.352** | **−0.213** |
| PGLS | | | | | | | | | | |
| Anthropoids (n = 27) | | | | | | | | | | |
| **bgPC1 vs. ln Vol** | **0.261** | **0.004** | **3.401** | **1.065** | **1.314** | **5.488** | **−3.719** | **1.376** | **−6.416** | **−1.022** |
| bgPC2 vs. ln Vol | 0.003 | 0.416 | 0.625 | 0.756 | −0.858 | 2.107 | −1.27 | 0.962 | −3.155 | 0.615 |
| bgPC3 vs. ln Vol | 0.063 | 0.111 | −0.727 | 0.440 | −1.589 | 0.135 | 0.879 | 0.570 | −0.238 | 1.996 |
| **ln VolSC vs. ln L** | **0.437** | **<0.001** | **0.502** | **0.109** | **0.288** | **0.716** | **3.144** | **0.124** | **2.901** | **3.388** |
| Hominids (n = 5) | | | | | | | | | | |
| bgPC1 vs. ln Vol | 0.221 | 0.240 | 4.440 | 3.042 | −1.521 | 10.402 | −4.627 | 4.499 | −13.445 | 4.192 |
| bgPC2 vs. ln Vol | 0.036 | 0.760 | −1.144 | 3.421 | −7.850 | 5.561 | 1.892 | 4.990 | −7.888 | 11.672 |
| bgPC3 vs. ln Vol | 0.062 | 0.687 | −0.392 | 0.883 | −2.122 | 1.338 | 0.596 | 1.309 | −1.970 | 3.162 |
| ln VolSC vs. ln L | 0.553 | 0.093 | 0.631 | 0.259 | 0.124 | 1.138 | 2.920 | 0.327 | 2.279 | 3.561 |
| Non-hominids (n = 22) | | | | | | | | | | |
| bgPC1 vs. ln Vol | 0.008 | 0.294 | 1.376 | 1.276 | −1.125 | 3.877 | −1.674 | 1.512 | −4.639 | 1.290 |
| bgPC2 vs. ln Vol | 0.030 | 0.445 | 0.763 | 0.978 | −1.154 | 2.679 | −1.444 | 1.152 | −3.701 | 0.814 |
| bgPC3 vs. ln Vol | 0.020 | 0.500 | −0.419 | 0.610 | −1.616 | 0.777 | 0.577 | 0.730 | −0.854 | 2.008 |
| **ln VolSC vs. ln L** | **0.504** | **<0.001** | **0.634** | **0.134** | **0.371** | **0.896** | **3.052** | **0.137** | **2.784** | **3.320** |

bgPC2 is slightly reduced as compared to bgPC1, but still significant and very high (λ = 0.9 and K = 1.08;).

The bgPC3 (17.4% of variance) is driven by both trajectory and size of the SCs, especially of the posterior one (*Figure 9e*), and displays less phylogenetic signal than bgPC1 and bgPC2 (λ = 0.93 and K = 0.53, *Table 7*). Along this axis *Homo*, *Ateles* and *Theropithecus* (*Figure 9b*) occupy the

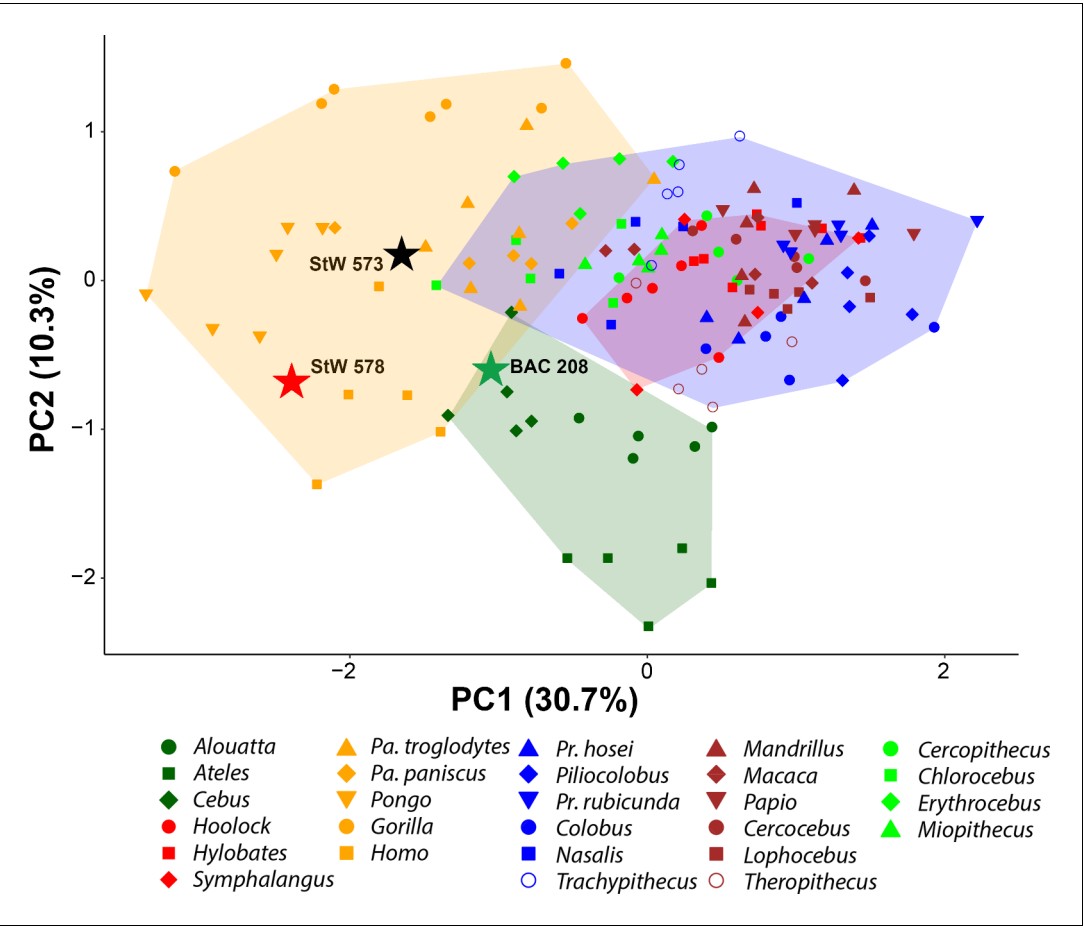

**Figure 8.** Tangent space of vestibular shape among the analyzed anthropoid sample as shown by bivariate plots of principal components from the PCA of deformation-based analysis. Variance explained by each PC is included within parentheses. Convex hulls correspond to: hominids (orange), hylobatids (red), cercopithecoids (blue), platyrrhines (green). Color code: dark green, platyrrhines; orange, hominids; red, hylobatids; brown, papionins; green, cercopithecins; blue, colobines. Hominids are distinguished from hylobatids and other anthropoids along PC1 (mostly negative vs. positive values, respectively), while PC2 tends to distinguish platyrrhines (more negative values) from catarrhines. Even if slightly more overlapping, the groups identified when observing the morphospace obtained with bgPCA are already present in the PCA.

The online version of this article includes the following source data for figure 8:

**Source data 1.** Individual scores for all the principal components (PC) yielded by the principal components analysis (PCA) of deformation-based 3DGM of vestibular shape for anthropoids.

positive end, as due to their large and anteriorly inclined posterior canal that protrudes laterally, an obtuse angle of the CC apex, large posterior canal, and an angle between the anterior and posterior canals that is close to 90°. Great apes and the majority of non-hominoid taxa fall in an intermediate position (*Figure 9b*), differing from the aforementioned genera by the less obtuse angle in the CC apex and a larger lateral canal. Hylobatids and, to a lesser extent, *Trachypithecus* show the lowest scores for bgPC3, as the result of the right to acute angle formed by the apex of the CC, more

**Table 6.** Probability of correct classification of individuals from the hierarchical clustering analyses of deformation fields according to the groups (hominids, hylobatids, cercopithecoids, platyrrhines) used in the bgPCA.

| Hominidae | Hylobatidae | Cercopithecoidae | Platyrrhini |
|---|---|---|---|
| 90.0% | 23.5% | 65.0% | 66.7% |

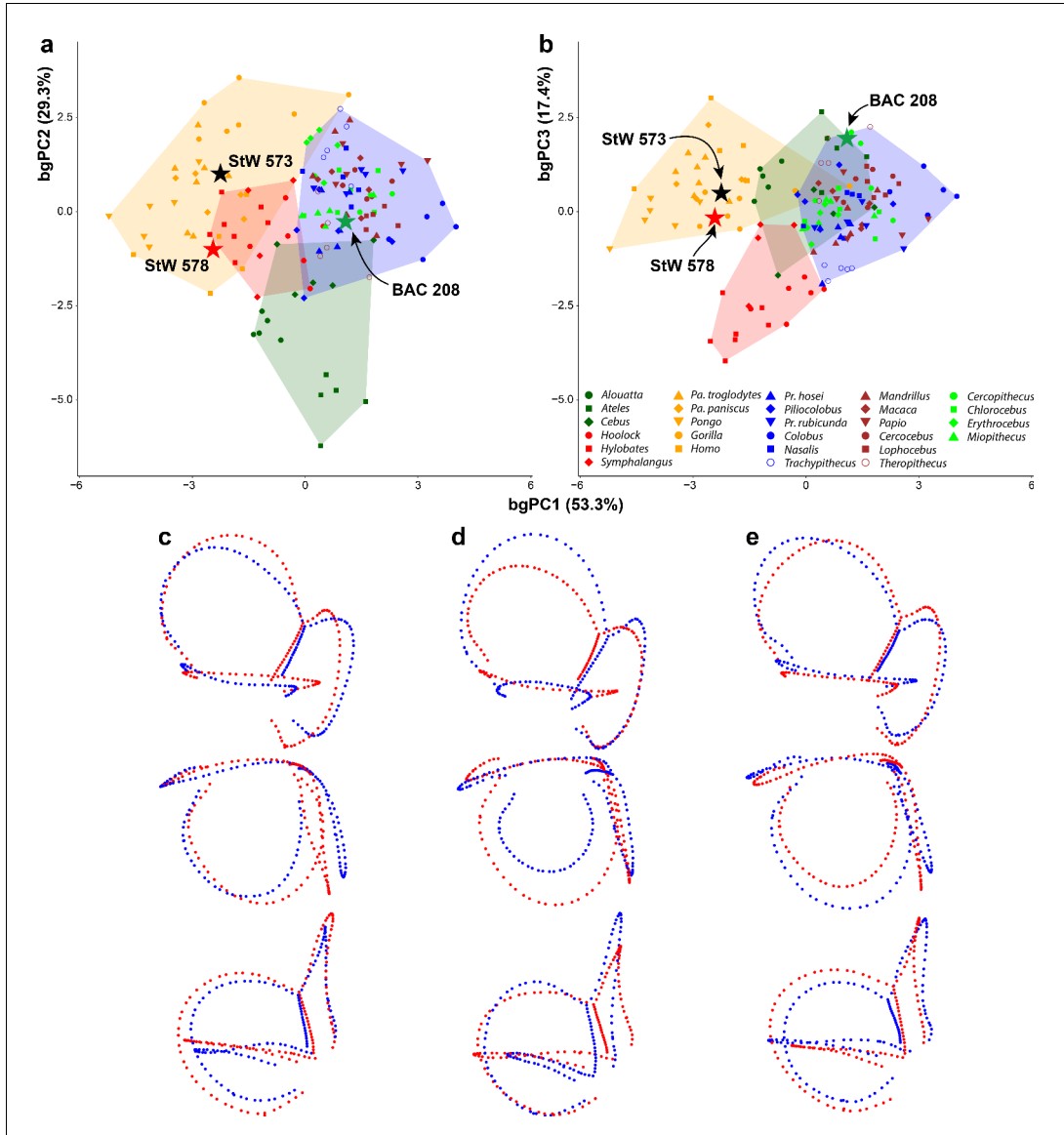

**Figure 9.** Main patterns of vestibular shape variation among the analyzed anthropoid sample as shown by bivariate plots of principal components from the landmark-based between-group principal components analysis (bgPCA) using major taxa as in *Figure 2*. (a) bgPC2 vs. bgPC1. (b) bgPC3 vs. bgPC1. Variance explained by each bgPC is included within parentheses. Lateral (top), superior (middle), and anterior (bottom) views of landmarks and semilandmark maximum (red) and minimum (blue) conformations for (c) bgPC1, (d) bgPC2, and (e) bgPC3. Color code as in *Figure 2*. Hominoids and cercopithecoids are distinguished along bgPC1 (mostly negative vs. positive values, respectively), which is mainly driven by the vertical compression of the anterior SC, as well as the position of the lateral (more anteriorly connecting with the vestibule) and the posterior (displaced posteromedially) canals. bgPC2, driven by the size of the anterior and posterior SCs relative to the lateral one, tends to distinguish platyrrhines (more negative values) from catarrhines, while bgPC3, driven by the size of the anterior and lateral SCs relative to that of the posterior canal, as well as by the relative orientation among the canals, tends to distinguish most hylobatids (more negative values) from great apes, humans, and cercopithecoids (more positive values). The two *Australopithecus* specimens fall within the hominid range, flanking that of hylobatids. Similarly to the deformation-based approach, StW 578 is more similar to humans (matching the distribution of *Homo*), while StW 573 is found within the scatter of points of *Pan* species. In the landmark-based analysis, *Oreopithecus* overlaps within cercopithecoids and also falls close to platyrrhines.

The online version of this article includes the following source data for figure 9:

**Source data 1.** Individual scores for all the principal components (bgPC) yielded by the between-group principal components analysis (bgPCA) of landmark-based 3DGM of vestibular shape for anthropoids, using major taxa (i.e., hominids, hylobatids, cercopithecoids, and platyrrhines) as grouping factor.

**Table 7.** Phylogenetic signal results for a between-group principal components analysis (bgPCA) applied to vestibular shape procrustes residuals in the analyzed sample of extant anthropoids.

|  | bgPC1 | bgPC2 | bgPC3 |
|---|---|---|---|
| Variance | 53.34% | 29.29% | 17.37% |
| Eigenvalue | 2.462 | 1.352 | 0.802 |
| Pagel's $\lambda$ | 1.000 (p<0.001) | 0.902 (p<0.001) | 0.932 (p=0.006) |
| Blomberg's $K$ | 1.226 (p=0.001) | 1.081 (p=0.001) | 0.528 (p=0.01) |

developed anterior and lateral canals relative to the posterior one, and a posteriorly tilted posterior canal.

When plotted onto the tangent space of extant taxa, the two *Australopithecus* specimens overlap with the range of great apes and humans for all the bgPCs (*Figure 9*, *Figure 9—source data 1*). They show a vertically compressed anterior canal together with well-separated lateral and posterior canals. The lateral canal is also moderately sinuous and its ampullar portion bends upwards, thus resulting in a negative score for bgPC1. The two specimens can be distinguished from one another by means of bgPC2 (*Figure 9a*), with StW 573 falling on positive values and StW 578 occupying negative ones. This is explained by the smaller lateral canal relative to the vertical SCs in StW 578, which therefore overlaps with extant human variation in both bgPC1 and bgPC2, whereas StW 573 overlaps instead with chimpanzees and bonobos (*Figure 9a*). Along bgPC3, both specimens display intermediate values due to the large posterior canal and for the almost right angle between the planes of the anterior and posterior SCs, overlapping with all extant anthropoids except hylobatids.

In turn, along bgPC1 *Oreopithecus* displays more positive values than hominoids and falls well within the range of non-hominoid anthropoids (it only slightly overlaps with the positive end of the hominoid distribution; *Figure 9*) due to its more coplanar lateral canal that almost intersects the plane defined by the posterior canal. Furthermore, due to its small lateral canal and fairly short CC, BAC 208 displays an intermediate value for bgPC2, within the range of extant catarrhines and slightly above the positive end of platyrrhine distribution (*Figure 9a*). Finally, along bgPC3 *Oreopithecus* falls on a positive score, differing from hylobatids, as a result of the acute angles found between anterior canal plane and those defined by both the posterior and lateral canals (*Figure 9b*).

Overall, the two 3DGM techniques used in this paper generally yielded similar results except for hylobatids and *Oreopithecus* (along bgPC1 alone). This is attributable to differences in the underlying methodological assumptions of each method when computing shape variation. In particular, our 3DGM landmark protocol measures the spatial trajectory of SCs based on their midline skeleton (*Gunz et al., 2012*) and hence it does not capture differences in volumetric proportions. In contrast, by comparing surfaces as a whole (*Durrleman et al., 2012b*; *Durrleman et al., 2012a*), the deformation analysis is particularly sensitive to volumetric differences. In addition, the amount of identified phylogenetic signal is very similar for both techniques, affecting the entire variance. Together with the results of the HCA, we confidently show that the separation found between the groups in the bgPCA of this study already exists in the shape data and that is not a spurious effect produced by the bgPCA method itself.

## Hominoid phylomorphospace of the vestibule

The phylomorphospace approach applied to vestibular shape variation in hominoids infers different branch lengths for hominids and hylobatids from the ancestral morphology estimated for crown hominoids, which falls much closer to great apes and humans than to hylobatids for both bgPC1 and bgPC3 (*Figure 10*). According to our reconstructions based on the extant taxa (*Figure 10*), the crown hominoid LCA vestibular morphology (*Figure 11a*) would be characterized by equally developed and slightly inflated SCs, a fairly vertically compressed anterior canal, and by the slender portion of the lateral canal connecting more anteromedially with the vestibule. The estimated morphology for the LCA of hylobatids (*Figure 11b*) resembles to some extent that of the crown hominoid LCA (slightly vertically compressed anterior canal and lack of intersection between the lateral and posterior canals), combined with more monkey-like features (markedly slender canals with inflated ampullae), and others exclusive to hylobatids (an obtuse angle between a slightly anteriorly

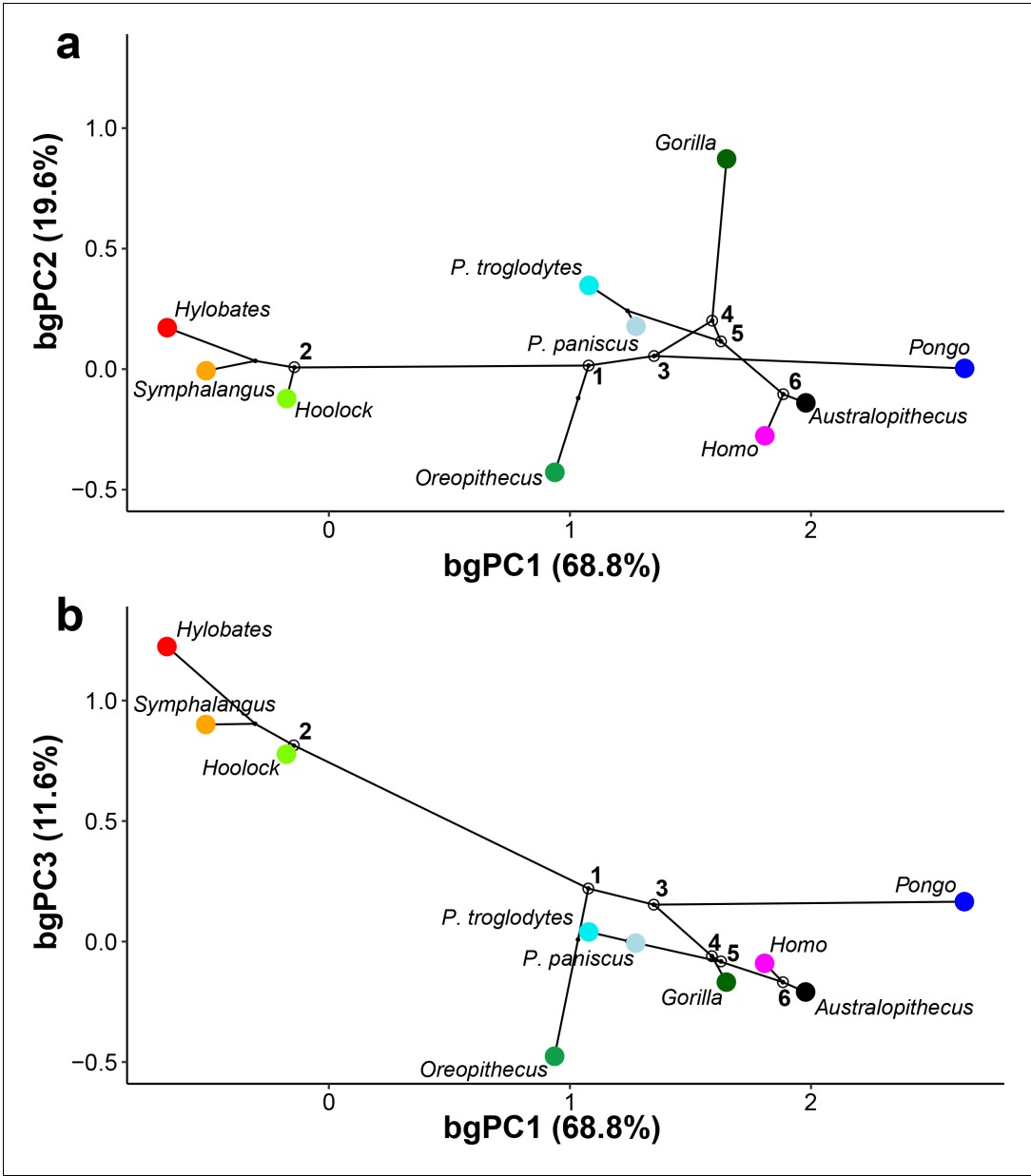

**Figure 10.** Reconstructed evolutionary history of the vestibular apparatus in the sample restricted to hominoids. The depicted phylomorphospaces are obtained by projecting the phylogeny displayed in *Figure 13* on the bivariate plots between principal components: (a) bgPC2 vs. bgPC1 (see *Figure 2a*); (b) bgPC3 vs. bgPC1 (see *Figure 2b*). Color code: red, *Hylobates lar*; orange, *Symphalangus syndactylus*; light green, *Hoolock hoolock*; blue, *Pongo pygmaeus*; dark green, *Gorilla gorilla*; cyan, *Pan troglodytes*; purple, *Pan paniscus*; fuchsia, *Homo sapiens*; chartreuse, *Oreopithecus bambolii*; black, *Australopithecus* sp. Key ancestral morphologies reconstructed using maximum likelihood for the last common ancestor of various clades are depicted as follows: 1, crown hominoids (hylobatids and hominids); 2, crown hylobatids (gibbons and siamangs); 3, crown hominids (great apes and humans); 4, crown hominines (African great apes and humans); 5, *Pan-Homo* clade; 6 *Australopithecus-Homo* clade.

protruding anterior canal and a small posterior canal relative to the others). Hylobatid genera are generally less diverging from one other than great apes and humans. *Hoolock* (*Figures 1f*, *3f*, *4f* and *5f*) apparently displays the most primitive morphology among hylobatids (with equally developed, rounded, almost orthogonal canals, and less anteriorly protruding anterior canal), while *Hylobates* (*Figures 1h*, *3h*, *4h* and *5h*) and, to a lesser extent, *Symphalangus* (*Figures 1g*,

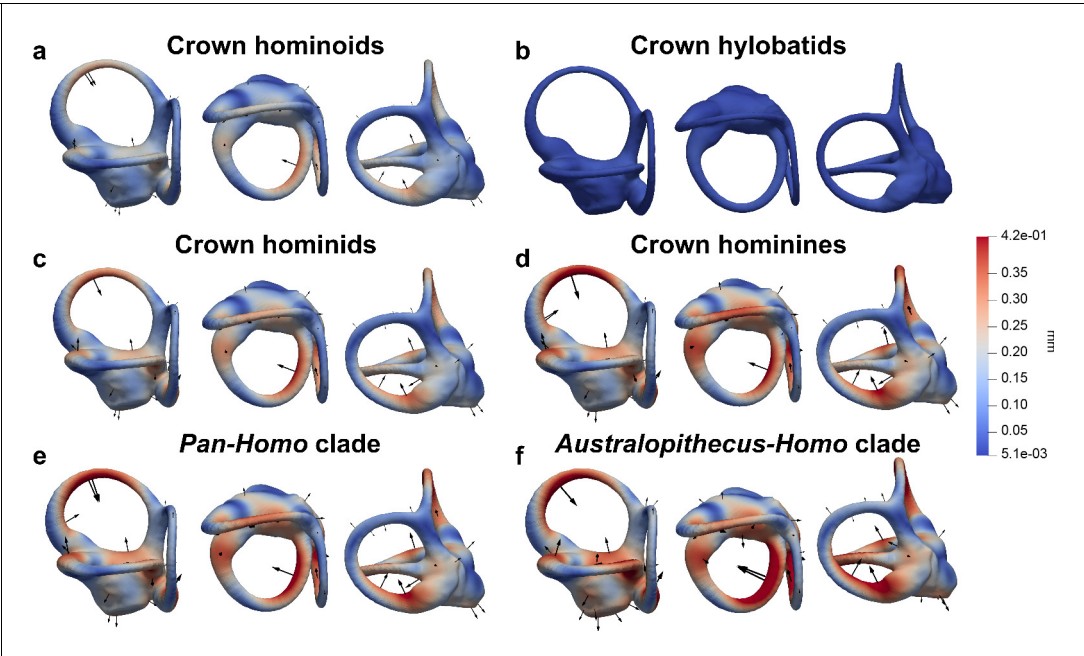

**Figure 11.** Reconstructed vestibular shape for the last common ancestors (LCA) of the main clades of interest as reconstructed using maximum likelihood methods for deformation-based 3DGM analyses applied to the anthropoid sample (*Figure 7*), in lateral (left), superior (middle), and posterior (right) views. Cumulative displacement variations are rendered by pseudocolor scale ranging from dark blue (<5.1 μm) to dark red (0.42 mm). Black arrows correspond to the vectors identifying the direction of displacement. The reconstructed LCAs depicted are the following: (**a**) crown hominoids; (**b**) crown hylobatids; (**c**) crown hominids; (**d**) crown hominines; (**e**) *Pan-Homo* clade; (**f**) *Australopithecus-Homo* clade.

*3g*, *4g* and *5g*) show the slenderest SCs and an extremely anteriorly protruding anterior canal (as noted by *Le Maître et al., 2017*).

The reconstructed morphologies for LCAs of crown hominids (*Figure 11c*) and, to a lesser extent, crown hominines (*Figure 11d*) and the *Pan-Homo* clade (*Figure 11e*) are not very far from the crown hominoid ancestral condition (*Figure 11a*) for any of the first three bgPCs. The crown hominid LCA is characterized by a lateral insertion of the slender portion of the lateral canal on the vestibule, a moderate medioventral displacement of the posterior canal, and an increased vertical compression of the anterior canal, in combination with thick and bulgy canals and well-developed vestibular recesses. The hominine and *Pan-Homo* clade LCAs possess stouter canals and are very similar to one another, distinguished only by the size of the lateral and anterior canals. The LCA of the *Pan-Homo* clade shows a larger and less vertically compressed anterior canal, and a smaller lateral one, which also connects more anteriorly with the vestibule. The hominin (*Australopithecus-Homo*) LCA (*Figure 11f*) is closer to *Homo*, being characterized by the stoutest volumetric proportions, with larger vertical canals relative to the *Pan-Homo* clade LCA, yet smaller than those found in humans. Its anterior canal is more vertically compressed than in *Homo*, rather resembling the morphology of *Pan*, while the posterior canal is rounded, thus being intermediate between the human (laterally projecting) and the chimpanzee (laterally compressed) morphology.

Among crown hominids, *Pan* (*Figures 1b–c*, *3b–c*, *4b–c* and *5b–c*) more closely resembles the morphology of the inferred LCAs of crown hominids and hominoids (*Figure 11a,c*) in the moderately inflated and equally developed SCs and in the degree of the vertical compression of the anterior canal. *Pongo* (*Figures 1e*, *3e*, *4e* and *5e*) occupies the positive end along bgPC1 (*Figure 10*) due to the possession of relatively small but extremely stout canals (especially the anterior one and the common crus). It also exhibits the most vertically compressed anterior canal and a 'triangular' lateral canal (i.e., showing straight slender portions of the bony labyrinth close to the ampulla and to the connection with the vestibule, as previously outlined by *Spoor and Zonneveld, 1998*). Finally, *Gorilla* and *Homo* (*Figures 1a*, *3d*, *4d* and *5d*) are derived in opposite directions along bgPC2 (*Figure 10a*), as the former exhibits increased lateral canal radius with a flattened cross-section,

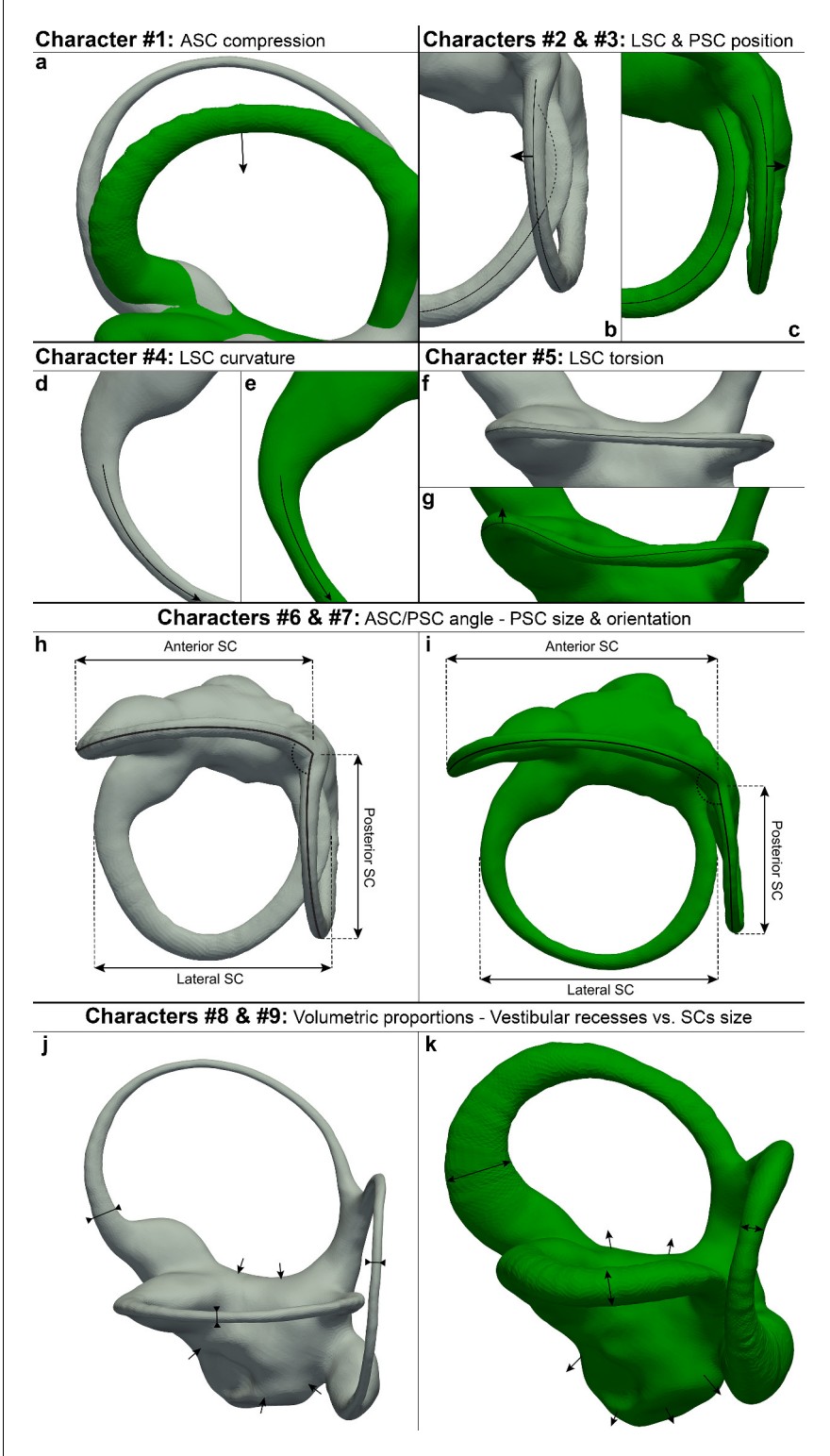

**Figure 12.** Illustration of hominoid, hominid and hylobatid synapormophies for the vestibular apparatus. See *Table 8* for further details. character #1: (**a**) rounded (gray) and compressed (green) anterior canal; character #2: (**b–c**) non-posteriorly displaced posterior canal (gray), posteriorly displaced posterior canal (green); character #3: (**b–c**) lateral canal intersecting(dashed line) the posterior one (gray), lateral canal non-intersecting the posterior one (in green); character #4: (**d-e**) curved (gray) and straight (green) medial portion of the lateral canal; character #5: (**f–g**) flat (gray) and bent upwards (green) trajectory of the lateral canal ampullar portion; character #6: (**h–i**) anterior
*Figure 12 continued on next page*

*Figure 12 continued*

and posterior SCs forming an angle (dotted arc) close to the right angle (gray), anterior and posterior canals forming an obtuse angle (green); character #7: (h–i) posterior canal equal in size to the other SCs and forming a right angle with the lateral canal (gray), small posterior canal relative the other SCs and inclined posteriorly respective to the lateral canal (green); character #8: (j–k) slender SCs (gray), stout canals (green); character #9: (j–k) small vestibular recesses (gray), enlarged vestibular recesses (green).

while humans retain quite rounded canals shape and cross-section, and show a reduction of the lateral canal, as opposed to more developed anterior and posterior canals.

*Australopithecus* is closer to humans than to any great ape, due to the possession of large vertical canals and stout volumetric proportions, being closest to the LCA of the *Australopithecus-Homo* clade and only slightly more derived in the same direction as *Pongo* due to the stouter canals (*Figure 10*). In contrast, *Oreopithecus* appears more plesiomorphic than humans and extant great apes in volumetric proportions, resembling those found in the LCA of crown hominoids. Nevertheless, *Oreopithecus* is clearly distinct from hylobatids in both SC stoutness and shape, being most clearly distinguished from gibbons and siamangs based on the acute angles defined the anterior canal with both the posterior and lateral canals (*Figure 10b*). *Oreopithecus* is also derived in terms of SC size (with the vertical canals much larger than the lateral one), similarly to humans albeit to a greater extent, and opposite to gorillas, due to the remarkably smaller lateral canal.

## Discussion

Previous research on the morphology of the vestibular apparatus among extant mammals has focused on its relationship to positional behavior (*Spoor et al., 2007*; *Perier et al., 2016*; *Le Maître et al., 2017*), particularly in order to make locomotor inferences in extinct species (*Spoor et al., 1994*; *Walker et al., 2008*; *Silcox et al., 2009*; *Ryan et al., 2012*). However, the phylogenetic signal embedded in vestibular morphology has not been adequately quantified among hominoids, because previous attempts were either exploratory (*Gunz et al., 2012*) or based on a restricted sample (*Le Maître et al., 2017*). Our results indicate that main anthropoid groups can be distinguished based on vestibular shape variation, and that there are important differences not only between hylobatids, great apes, and humans, but also among extant great ape genera. A significant phylogenetic signal is found to affect the entire variance of the anthropoid sample. Thus, the shape of the SCs is overall informative from a phylogenetic viewpoint—as hypothesized for strepsirrhine primates (*Lebrun et al., 2010*) and carnivorans (*Schwab et al., 2019*), but in contrast to previous results for hominoids (*Le Maître et al., 2017*) and some other mammals (*Grohé et al., 2016*; *Costeur et al., 2018*).

Based on the analysis of the shape of the vestibular apparatus, we identify several potential hominoid synapomorphies (*Table 8*; *Figure 12a–g*), including among others a posteromedially displaced posterior canal and a straight segment between the lateral-most point of the lateral canal and its anteromedially situated insertion on the vestibule. These features result in an anteromedially located lateral canal (i.e., the plane defined by the posterior canal is always separated from the trajectory of the lateral canal, even when the latter is well developed, as in *Gorilla* and *Hylobates*). This would imply an increased sensitivity for angular accelerations occurring along the coronal plane, which has been correlated with orthogrady in extant hominoids (*Le Maître et al., 2017*).

The most evident character shared by extant apes and humans (even if somewhat variable in the latter and in *Hoolock*), and further displayed by the extinct genera analyzed here, is the vertical compression of the anterior canal (*Figure 12a*), as noted for great apes only in a previous analysis (*Spoor and Zonneveld, 1998*). Since the subarcuate fossa arguably constrains the shape of the anterior canal (*Jeffery et al., 2008*), the hominoid morphology might be related to the absence of the fossa in great apes and siamangs (*Moyà-Solà and Köhler, 1993*; *Gannon et al., 1988*; *Spoor and Leakey, 1996*). However, the combination of a well-developed fossa and marked vertical compression of the anterior SC found in *Hylobates* argues against this hypothesis. The latter is further rejected by the cercopithecoid morphology, characterized by a rounded anterior SC even in the largest terrestrial genera (*Papio*, *Theropithecus*, and *Mandrillus*), which unlike other cercopithecoids

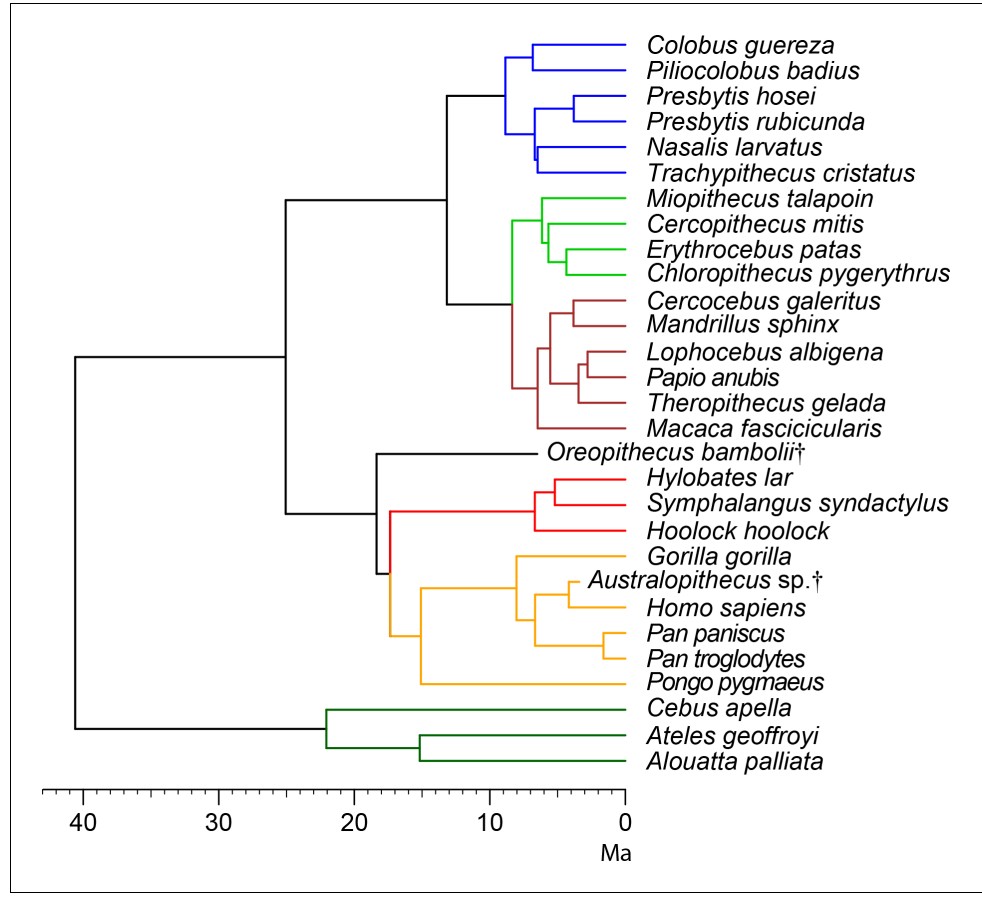

**Figure 13.** Time-calibrated molecular phylogeny of extant anthropoids used in the analyses of phylogenetic signal and PGLS regressions as inferred from a species supermatrix. Fossil taxa have been added a posteriori according to their phylogenetic position and their point estimates correspond to their last occurrence in the fossil record (*Rook et al., 2000*; *Wood and Boyle, 2016*). *Oreopithecus* is here considered as a stem hominoid, predating the split between hominids and hylobatids, while *Australopithecus* has been added following the first appearance datum for *A. africanus* based on the Jacovec Cavern specimens. For the phylomorphospace the phylogenetic tree has been pruned to include extant and extinct hominoids only. Branches are color-coded: dark green, platyrrhines; brown, papionins; green, cercopithecins; blue, colobines; orange, hominids; red, hylobatids. Extinct taxa are denoted by a dagger.

display a much reduced (or even absent) subarcuate fossa (*Gannon et al., 1988*; *Spoor and Leakey, 1996*).

As a result of vertical compression, all hominoids display the anterior canal projected anterosuperiorly to some extent (less accentuated in *Homo* and *Hoolock*). Such a projection of the anterior canal in *Hylobates* has been interpreted as a hylobatid synapomorphy (*Le Maître et al., 2017*). Our results raise doubts about the latter view and indicate instead that the anterior projection of this canal is variable within both hylobatids and hominids, and that the LCA of crown hylobatids might have not shown the patent elongation of the anterior canal that is found in *Hylobates* and *Symphalangus*. On the other hand, here we identify two potential synapomorphies for hylobatids (*Table 8*; *Figure 12h,i*): an obtuse angle between the planes defined by the anterior and posterior canals, and a posteriorly-inclined posterior canal, which is smaller relative to the anterior and lateral ones.

In turn, bgPC1 clearly separates hominids (great apes and humans) from cercopithecoids and hylobatids and enables the identification of some potential hominid synapomorphies (*Table 8*; *Figure 12j,k*). In particular, hominids differ from other anthropoids, including hylobatids, by derived volumetric proportions of the SCs (stouter canals relative to their length, or shorter canals relative to their volume) even when size-scaling considerations are taken into account, as well as by the possession of more extensive vestibular recess for a similar size of the SCs (*Table 8*; *Figure 12j,k*), as

**Table 8.** Phylogenetically informative discrete characters of vestibular morphology that represent potential synapomorphies of either hominoids or hominids.

| Character # | Character definition | Character states[a] | Synapomorphic for |
|---|---|---|---|
| #1 | Anterior SC[b] | 0 = rounded; 1 = vertically compressed | Hominoidea |
| #2 | Posterior SC | 0 = non posteromedially displaced; 1 = posteromedially displaced | Hominoidea |
| #3 | Insertion of the lateral SC slender portion on the vestibule | 0 = posterior (lateral SC intersecting the posterior SC plane); 1 = anterior (not intersecting) | Hominoidea |
| #4 | Lateral SC medial portion | 0 = curved; 1 = straight | Hominoidea |
| #5 | Trajectory of the lateral SC ampullar portion | 0 = flat; 1 = bent upwards | Hominoidea |
| #6 | Angle between the planes identified by the anterior and posterior SCs | 0 = close to right angle, 1 = obtuse | Hylobatidae |
| #7 | Inclination and size of the posterior SC relative to the size of the anterior and lateral SCs | 0 = forming a right angle and equal or larger in size, 1 = posteriorly tilted and smaller | Hylobatidae |
| #8 | Robusticity of SCs | 0 = slender; 1 = stout | Hominidae |
| #9 | Extension of vestibular recesses relative to that of the SCs | 0 = smaller; 1 = similar in size | Hominidae |

Abbreviations: SC = semicircular canals.

[a] Character state 0 represents the primitive condition reconstructed for the last common ancestor of crown catarrhines.

[b] Some platyrrhines display a superior eccentricity of the anterior SC that might be apomorphic.

reflected in bgPC1. In particular, cercopithecoids and hylobatids completely overlap due to the possession of slender SCs, while hominids as a whole (even if more markedly orangutans) differ by their swollen and relatively shorter SCs (with only few cercopithecins falling within the hominid range). This might be related to the fact that hylobatids and cercopithecoids, unlike great apes, are swift moving animals that make fast and large head movements, thus requiring a limited duct sensitivity to avoid overstimulation and a quick response to angular displacement (*Spoor and Zonneveld, 1998*). This hypothesis is supported by biophysical models suggesting that the length of the membranous ducts is inversely proportional to their sensitivity and that a larger lumen of the ducts correlates with a reduced steadiness of the response to an external angular stimulus (i.e., the abrupt change of the position and/or posture) (*Muller, 1994*; *Rabbitt et al., 2004*). Therefore, species with shorter and thicker ducts (such as hominids) require more time to perceive and react to sudden head displacements, while being more sensitive to fine movements. Nevertheless, caution must be used when inferring the lumen of the ducts from that of the bony canals, as the amount of the SC cross-section occupied by perilymphatic space is variable depending on the species (*Ramprashad et al., 1984*; *Spoor and Zonneveld, 1998*).

Superimposed on the aforementioned hominoid and hominid synapomorphies (*Table 8*; *Figure 12*), there are also marked differences among hominid genera. Such differences mainly relate the relative size among the SCs (which varies particularly along bgPC2 and bgPC3), while hylobatid genera are less diverse in this regard. Chimpanzees and bonobos are characterized by equally developed SCs and a moderately short CC. Similarly, orangutans possess evenly proportioned SCs and can be distinguished from *Pan* by a shorter CC, more inflated canals, and a greater vertical compression of the anterior SC. Gorillas display the largest intrageneric variability in the studied sample, especially with regard to SCs slenderness, coupled with some other distinctive traits (obtuse angle of the CC apex, and longer lateral SC and CC).

Humans differ from apes in the enlarged vertical canals, a laterally protruding and inferiorly displaced posterior canal, and a moderately smaller lateral canal. Relatively enlarged vertical canals are also found in *Theropithecus*. The human morphology has been linked to bipedalism (*Spoor et al., 1994*; *Spoor et al., 2003*), as accelerations during bipedal walking mainly occur along the vertical axis, so that the broadly similar morphology of *Theropithecus* might be related to the bipedal

**Table 9.** Sample of extant anthropoid specimens analyzed in this paper based on µCT image stacks. See **Supplementary file 1** for further details on each specimen.

| Family | Species | n | M | F | ? |
|---|---|---|---|---|---|
| Atelidae | *Alouatta palliata* | 5 | 3 | 2 | 0 |
| Atelidae | *Ateles geoffroyi* | 5 | 1 | 4 | 0 |
| Cebidae | *Cebus apella* | 5 | 3 | 2 | 0 |
| Cercopithecidae | *Cercocebus galeritus* | 5 | 3 | 2 | 0 |
| Cercopithecidae | *Cercopithecus mitis* | 5 | 0 | 5 | 0 |
| Cercopithecidae | *Chlorocebus pygerythrus* | 5 | 2 | 3 | 0 |
| Cercopithecidae | *Colobus guereza* | 5 | 2 | 3 | 0 |
| Cercopithecidae | *Erythrocebus patas* | 5 | 3 | 2 | 0 |
| Cercopithecidae | *Lophocebus albigena* | 5 | 2 | 3 | 0 |
| Cercopithecidae | *Macaca fascicularis* | 5 | 1 | 4 | 0 |
| Cercopithecidae | *Mandrillus sphinx* | 5 | 5 | 0 | 0 |
| Cercopithecidae | *Miopithecus talapoin* | 5 | 3 | 2 | 0 |
| Cercopithecidae | *Nasalis larvatus* | 5 | 0 | 5 | 0 |
| Cercopithecidae | *Papio anubis* | 5 | 3 | 2 | 0 |
| Cercopithecidae | *Piliocolobus badius* | 5 | 4 | 1 | 0 |
| Cercopithecidae | *Presbytis hosei* | 5 | 1 | 4 | 0 |
| Cercopithecidae | *Presbytis rubicunda* | 5 | 2 | 3 | 0 |
| Cercopithecidae | *Theropithecus gelada* | 5 | 4 | 1 | 0 |
| Cercopithecidae | *Trachypithecus cristatus* | 5 | 0 | 5 | 0 |
| Hylobatidae | *Hoolock hoolock* | 6 | 2 | 4 | 0 |
| Hylobatidae | *Hylobates lar* | 7 | 0 | 7 | 0 |
| Hylobatidae | *Symphalangus syndactylus* | 4 | 2 | 2 | 0 |
| Hominidae | *Gorilla gorilla* | 7 | 2 | 5 | 0 |
| Hominidae | *Homo sapiens* | 5 | 2 | 3 | 0 |
| Hominidae | *Pan paniscus* | 5 | 1 | 4 | 0 |
| Hominidae | *Pan troglodytes* | 7 | 4 | 3 | 0 |
| Hominidae | *Pongo pygmaeus* | 6 | 0 | 4 | 2 |

Abbreviations: n, total number of specimens; M, males; F, females; ?, unknown sex.

shuffling characteristic of this genus during foraging, causing them spend an extremely large amount of time with an erect trunk posture (**Wrangham, 1980**).

Irrespective of the functional implications of the variation in vestibular morphology among anthropoids, our analyses show that this variation bears strong phylogenetic signal and, hence, has potential for reconstructing the evolutionary history of the group—particularly hominoids, which in spite of their extant decimated diversity are more variable in this regard (particularly when size differences among the SCs are considered) (**Spoor and Zonneveld, 1998**; **Le Maître et al., 2017**) than the taxonomically more diverse cercopithecoids. Although functional demands frequently lead to the independent evolution of similar morphologies (homoplasy), often function is not decoupled from—but superimposed on—phylogeny, with many clades being characterized by synapomorphic features linked to the adaptation for new functions. Therefore, to the extent that vestibular morphology appears to be linked to positional behavior (**Spoor et al., 1994**; **Spoor et al., 2007**; **Walker et al., 2008**; **Silcox et al., 2009**; **Ryan et al., 2012**; **Le Maître et al., 2017**), the higher variation of vestibular morphology displayed by hominoids compared to cercopithecoids agrees with the more diverse and varied locomotor repertoires of the former. This is because cercopithecoids as a whole are largely pronograde terrestrial quadrupeds that mostly differ in the degree of arboreality vs. terrestriality (**Fleagle, 2013**; **Gosselin-Ildari, 2013**), while crown hominoids are characterized by a derived

and versatile orthograde body plan and associated adaptations throughout the body that enable very different and very specialized antipronograde behaviors—vertical climbing (all apes), ricochetal brachiation (hylobatids), arboreal quadrumanous suspension and clambering (orangutans), below-branch arm-swinging as well as semiterrestrial knuckle-walking (African apes), and terrestrial bipedalism (humans) (**Hunt, 1991**; **Thorpe and Crompton, 2006**).

The fact that, in terms of positional behavior, extant hominoid lineages have more significantly diverged in different directions from their last common ancestor with cercopithecoids explains why extant hominoids more strongly differ in vestibular features—even if the functional link of some vestibular features remains to be better determined. Determining the order in which these features evolved is therefore required to use them for inferring the phylogenetic placement of extinct hominoids. Besides proposing various potential synapomorphies for the hominoid and hominid clades, we further reconstruct the evolution of the vestibular apparatus in this group by estimating ancestral vestibular morphotypes by means of maximum likelihood and a molecular phylogeny. From the LCA of crown hominoids, great apes and humans appear derived in the opposite direction of hylobatids with regard to their volumetric proportions (stout vs. slender SCs, respectively). According to our ancestral state reconstruction for crown hominoids, in this regard hylobatids appear secondarily convergent with cercopithecoids. However, this hypothesis (and the alternate one, that hylobatids merely reflect more closely the primitive condition for crown catarrhines as a whole) should be tested by means of adding extinct stem cercopithecoids and fossil hominoids of less controversial affinities than *Oreopithecus* to the analysis. It is noteworthy that *Hoolock*, in agreement with the basal position of this genus among extant hylobatids, apparently retains a more primitive morphology than other hominoids for various features. While the characters related to an anteromedial displacement of the lateral SC appear synapomorphic for hominoids, the only incipient vertical compression of the anterior canal SC morphology of *Hoolock* might be plesiomorphic for hylobatids, in which case the marked vertical compression of this canal would be a synapomorphy of hominids only, with the remaining hylobatid genera having also evolved it in parallel.

While various great ape lineages and humans further diverged from one another from the more derived condition of the reconstructed crown hominoid LCA, as noted above gibbons and siamangs might have secondarily converged to some extent with cercopithecoids by evolving slenderer SCs, presumably as a result of similar evolutionary pressures posed by fast-moving types of locomotion. Based on our reconstructed ancestral morphotypes, bonobos and (to a lesser extent) chimpanzees would be closer to the LCAs of hominids, hominines, and the *Pan-Homo* clade, than either humans or the remaining extant great apes (gorillas and orangutans). The latter would have diverged from the hominid LCA in markedly different directions both in terms of SC configuration and stoutness. Our results therefore support the view that not only hominins, but also gorillas and, to a large extent, the orangutan lineage diverged from ancestors with a largely *Pan*-like vestibular morphology. It would be tempting to interpret this pattern in locomotor terms (e.g., by suggesting a semiterrestrial ancestry not only for hominines, but also for crown hominids as a whole). However, caution is required as other selection pressures and/or non-adaptive factors could have potentially played an equally, if not more significant, role in determining vestibular shape variation in this group.

With regard to the fossil hominoids analyzed here, *Australopithecus* not only displays the various hominoid synapomorphies mentioned above, but also hominid-like volumetric proportions of the SCs and, as expected in a bipedal hominin, human-like vestibular features such as large anterior and posterior canals. This is in agreement with the large amount of habitual bipedal behaviors inferred for the almost complete skeleton (StW 573) to which one of the analyzed specimens belongs (**Heaton et al., 2019**), as well as previous analyses of the inner ear as a whole (**Beaudet et al., 2019b**). Following our ancestral state reconstruction, *Australopithecus* appears derived in the same direction as humans, although it more closely resembles the morphology of the *Australopithecus-Homo* clade LCA, which is derived from the reconstructed LCA of the *Pan-Homo* clade in the opposite direction as chimpanzees and bonobos are. The fact that the two analyzed specimens of *Australopithecus* are classified as humans with a moderately high probability reflects the fact that their vestibular morphology already approximates the human condition, although maintaining plesiomorphic characters, particularly in the specimen that more closely resembles chimpanzees. On the other hand, their similar classification probability with other extant great ape genera is consistent with a more primitive vestibular morphology. It is noteworthy that, although the two analyzed individuals are very similar to one another (**Beaudet et al., 2019b**), they display noticeable differences in

vestibular morphology, with StW 578 showing more human-like (even if stouter, approximating the orangutan morphology) canals and StW 573 retaining a more African great ape-like morphology. This could be related with diachronic changes within South African *Australopithecus* or to its previously noted heterogeneity (*Clarke, 2013*; *Grine, 2013*), and might help to discern, coupled with other features, the number of species represented among the current samples.

Unlike *Australopithecus*, *Oreopithecus* displays a mosaic vestibular morphology that defies a simple phylogenetic interpretation, as it does not fit well among the variation displayed by any extant hominoid genus. This is due to the combination of some hominoid and hominid synapomorphies with more plesiomorphic, cercopithecoid-like, or even platyrrhine-like, features. In particular, the hominid-like volumetric proportions of the *Oreopithecus* SCs would support the contention that this taxon is a great ape (*Begun et al., 1997*), or even a member of the European dryopithecine radiation, as previously argued by some authors (*Köhler and Moyà-Solà, 1997*; *Harrison and Rook, 1997*). However, other, apparently more plesiomorphic vestibular features, are at odds with such an interpretation, and even with the previous suggestion that *Oreopithecus* postcranium would be consistent with a giant hylobatid that emphasized cautious climbing (*Sarmiento, 1987*). In particular, the vestibular morphology of *Oreopithecus* does not overlap with that of extant hylobatids in any respect, particularly differing by the acute angle between the anterior and posterior canal planes and by the large posterior canal relative to the others. *Oreopithecus* also appears more primitive than crown hominoids in the shape of the lateral canal, which is flat (rather than displaying an upwards bent ampullar portion) and posteriorly displaced (especially in the junction between the ampulla and the vestibular recesses). In these regard, the *Oreopithecus* morphology more closely resembles that of cercopithecoids and platyrrhines, respectively, possibly reflecting a plesiomorphic condition that would be more consistent with a stem hominoid status, as recently supported by some other authors (*Nengo et al., 2017*).

If our interpretation above is correct, then the stout volumetric proportions of *Oreopithecus* would be homoplastic with those of great apes and humans, representing an independent acquisition that might be functionally related to the evolution of a slower mode of locomotion—in agreement with previous analysis of the inner ear of this taxon (*Rook et al., 2004*; *Ryan et al., 2012*) and the possession of an orthograde body plan with adaptations related to cautious vertical climbing and forelimb-dominated suspension (*Sarmiento, 1987*; *Harrison and Rook, 1997*). This is plausible given that hylobatids appear to some extent convergent in this regard to cercopithecoids, due to their agile locomotion. This suggests that volumetric proportions are quite labile in evolutionary terms, so that other features and functional considerations must also be considered when interpreting the vestibular morphology of extinct taxa in phylogenetic terms. Interestingly, *Oreopithecus* resembles australopiths and humans in the possession of a larger vertical and posterior canals relative to the lateral one, which apparently represents another homoplasy that would lend some support to the controversial claim that bipedalism featured prominently among the locomotor repertoire of this taxon (*Köhler and Moyà-Solà, 1997*; *Rook et al., 1999*). However, the SCs of *Oreopithecus* are clearly distinguishable from those present in *Homo* and *Australopithecus* regarding volumetric proportions, orientation, and shape. This rules out a hominin-like bipedalism for *Oreopithecus*—in further agreement with the lack of the lower torso features than in australopiths and humans are functionally linked to committed bipedalism (*Hammond et al., 2020*)—but would not be at odds with the possession of more varied orthograde positional behaviors combining climbing with a different type of bipedalism (more related to a stable bipedal stance and short distance shuffling instead of fast walking or running), as previously inferred based on the foot of this taxon (*Köhler and Moyà-Solà, 1997*).

In conclusion, our study provides new insight into the evolution of the vestibular apparatus in hominoids and confirms the potential of SC shape for investigating further the phylogenetic affinities of fossil apes, which are still controversial due to the inherent limitations of the fossil record. This is not to say that functional considerations must not be taken into account—rather the contrary, as several of the discussed vestibular features are arguably linked with the demands of particular positional behaviors, as noted by previous authors (*Spoor et al., 2007*; *Walker et al., 2008*; *Silcox et al., 2009*; *Ryan et al., 2012*; *Perier et al., 2016*; *Le Maître et al., 2017*). However, as exemplified by the analysis of the extinct hominin *Australopithecus*, the various characters identified as potentially synapomorphic for either crown hominoids or hominids offer the prospect of refining the phylogenetic placement of fossil apes for which their stem vs. crown hominoid status is controversial—as

these features can be easily scored from CTs of the petrosal bone and incorporated into formal cladistic analyses including information from other anatomical areas. On the other hand, our ancestral state reconstructions rely mainly on living taxa, which is potentially problematic in the case of hominoids, which were much more diverse in the Miocene and appear quite prone to homoplasy, particularly with regard to the locomotor adaptations of the few surviving lineages. Even if the quantification of phylogenetic signal based on the phylogeny of extant taxa indicates that vestibular morphology overall is not significantly affected by homoplasy, the evolutionary history of vestibular morphology presented here on the basis of ancestral morphotypes should be treated with caution as a set of working hypotheses that require further testing based on the information provided by a larger fossil sample. In particular, given the relationship between vestibular morphology and positional behavior, and the fact that the locomotor apparatus of extinct hominoids frequently displays a mosaic of primitive and derived features unknown among the surviving lineages (e.g., *Moyà-Solà et al., 2004*; *Alba, 2012*; *Alba et al., 2015*; *Böhme et al., 2019*), it may be predicted that the vestibular morphology of extinct hominoids will similarly display unique combinations of features. This is illustrated here by the condition of *Oreopithecus*, which is nevertheless most consistent with that of a stem hominoid somewhat convergent with hominids in terms of locomotion. In any case, our conclusions should be subjected to further scrutiny in the future by means of the inclusion of additional fossil taxa, with emphasis on Miocene hominoids as well as stem cercopithecoids.

## Materials and methods

### Sample composition and acquisition

The analyzed sample includes microcomputed tomography (μCT) scans of 142 dried anthropoid crania belonging to 27 species and 25 genera, including all extant great ape genera and a selection of hylobatids, cercopithecoids, and platyrrhines (*Table 9* and *Supplementary file 1*). A few specimens are juveniles instead of adults, but this should not affect their vestibular morphology as the bony labyrinth ossifies in early prenatal stages, bounding its shape and size (*Jeffery and Spoor, 2004*; *Perier et al., 2016*). The hominoid subsample consists of 48 individuals belonging to 8 species and seven genera (*Supplementary file 1*). For each specimen, the bony labyrinth was virtually extracted (from the left side when possible, or otherwise from the right side and mirrored) by segmenting the μCT image stacks (voxel size reported in *Supplementary file 1*). Virtual 3D models were generated using Avizo 9.0.1 software (FEI Visualization Sciences Group). The fossil sample consists of one left bony labyrinth belonging to the late Miocene stem-hominoid *Oreopithecus bambolii* (*Rook et al., 2004*) and of two right inner ear that have been virtually extracted from the *Australopithecus* specimens StW 573 and StW 578 from Sterkfontein (*Beaudet et al., 2019b*).The vestibular apparatus was separated from the cochlea by cutting right under the oval window and the saccule, and filling the resulting hole with a flat surface in Geomagic Studio 2014 software (3D Systems).

### Shape analysis

Our 3DGM approach is based on deformation methods (*Durrleman et al., 2012b*; *Durrleman et al., 2012a*), which do not rely on a priori defined landmarks but consider instead the geometrical correspondences between continuous surfaces, and are particularly convenient for comparing overall shape and complex 3D surface changes (*Durrleman et al., 2012b*; *Dumoncel et al., 2014*; *Beaudet et al., 2016a*). This method relies on the construction of a sample-average surface model (template) and its deformation to the investigated surfaces (*Durrleman et al., 2012b*; *Durrleman et al., 2012a*; *Beaudet et al., 2016b*). Unlike in classical landmark-based 3DGM analyses, the surfaces are represented by a set of oriented faces and the comparisons do not assume a point-to-point correspondence between samples (*Durrleman et al., 2012b*). Prior to the analysis, the unscaled vestibular surfaces were aligned and scaled using the 'Align Surface' module of Avizo 9.0. Then, deformations between surfaces were mathematically modeled as a diffeomorphism (i.e., a one-to-one deformation of the 3D space that is smooth, invertible, and with a smooth inverse), and a set of momenta (vectors representing the flow of deformations from the initial position of the control points on the template to the target shape) were estimated with Deformetrica 3 software. Due to its high-demanding computational power, analyses were run in the CALMIP supercomputing center (Toulouse, France).

We inspected interspecific major patterns of shape variation by means of between-group principal components analysis (bgPCA) of the deformation-based shape residuals, using major clades (i.e., platyrrhines, cercopithecoids, hylobatids and hominids) as grouping factor. The restricted platyrrhine sample included in this study is aimed to serve as an outgroup for catarrhines, since the description of the vestibular morphology variation among New World monkeys as a whole is beyond the scope of this paper. Each group has been designed to include a large number of individuals (>>10) in order to prevent spurious separation between the groups used in the analysis. We preferred bgPCA over linear discriminant analysis (LDA) because the latter produces overexaggerated separation among groups when the number of variables is close to the number of the analyzed individuals (*Mitteroecker and Bookstein, 2011*). Taking into account that some authors have recently recommended caution when interpreting bgPCA results, as they might present spurious grouping (*Cardini et al., 2019*), we compared our results with those of a landmark-based 3DGM analysis ran on the same sample and investigated the presence of a preexisting group structure (See '*Exploration of a preexisting group structure in the tangent space of the vestibular shape*' section for further details).

Correlation between SC shape and size (allometry) was assessed using multivariate regression of the deformation fields against log-transformed cube root of the entire vestibular volume (ln Vol, in mm$^3$), as well as bivariate regressions computed for each bgPC against ln Vol. We further inspected the correlation between log-transformed cube root of the SC volume (ln VolSC, in mm$^3$; including the SCs and the CC cut at their connection with the vestibular recesses) and log-transformed length (ln L, in mm; measured along the streamline of the SCs and of the CC). All these regressions were performed for the whole anthropoid sample, as well as for hominids and non-hominids separately, as we detected several differences for the former. In the case of the ln VolSC vs. ln L regression, we also checked the homogeneity of slopes and intercepts between hominoids and the rest of the sample via analysis of covariance (ANCOVA). Regressions were performed by means of ordinary least-square linear regression (OLS) as well as by taking into account the phylogenetic non-independence between data, that is by fitting our linear models via phylogenetic generalized least squares (PGLS). The data used to compute the regressions are given in *Figure 7—source data 1*.

The hierarchical clustering analysis (HCA) of the deformation fields (used to identify preexisting group structure embedded in our shape data) has been computed *caret* v6.0–84 (*Kuhn, 2008*) and *Factominer* v1.34 (*Le et al., 2008*) R packages. The discrimination and the amount of overlap among the groups defined a priori for the bgPCA has been quantified by computing the number of correctly classified individuals with cross-validation using the *Morpho* v2.6 (*Schlager, 2014*) R package. Finally, we estimated the posterior probabilities of group membership for the fossil specimens based on the Mahalanobis distances between their bgPC scores and group centroids using the '*typprobClass*' function of *Morpho* v2.6 (*Schlager, 2014*) R package. These probabilities, which were computed for both bgPCA (based on main anthropoid groups as well as on extant hominoid genera only) denote the likelihood of the specimens to belong to each group without assuming that it must belong to one of them (which is required when comparing the fossils with extant genera), so that the sum of probabilities for each specimen does not equal one. Probabilities < 0.05 indicate that the specimen falls outside the variability of that particular group. Further statistical analyses were carried out using different R packages in RStudio v.1.1.453 for R v.3.5.0: *ape* v5.1 (*Paradis, 2012*), *phytools* v0.6–60 (*Revell, 2012*), *Morpho* v2.6 (*Schlager, 2014*), *caper* v1.0.2 (*Orme et al., 2013*), and *geomorph* v3.1.1 (*Adams et al., 2019*).

## Phylogenetic signal analysis

To assess phylogenetic signal, that is the degree to which related species resemble each other (*Felsenstein, 1985*; *Harvey and Pagel, 1991*), we used a phylogenetic tree (*Figure 13*) derived from a time-calibrated molecular phylogeny for the extant taxa (*Springer et al., 2012*). Estimated mean divergence dates for the various included extant clades are indicated in *Figure 13* (see *Table 1* and S1 in *Springer et al. (2012)* for 95% composite credibility intervals). The phylogenetic placement of *Oreopithecus* has been controversial; for our purposes here, we followed *Nengo et al. (2017)* in considering this taxon as a stem-hominoid, although other possibilities are discussed in the text. The node of the *Oreopithecus*-crown hominoid divergence has been placed 1 Ma older than the divergence of crown apes and humans and its tip corresponds to its last occurrence in the fossil record (7.0–6.5 Ma; *Rook et al., 2000*). For the South African *Australopithecus* sp., we used the published

first appearance datum for *Australopithecus africanus* (4.02 Ma) that includes the Jacovec specimens into the taxon (*Wood and Boyle, 2016*).

We computed Pagel's λ (*Pagel, 1999*), Blomberg's *K* (*Blomberg et al., 2003*) and $K_{mult}$ (*Adams, 2014*) using *phytools* v.0.6–44 (*Revell, 2012*) and *geomorph* v3.1.1 (*Adams et al., 2019*) R packages. These metrics compare the variance in the phylogenetic tree tips relative to that expected under a Brownian motion evolutionary model. Pagel's λ is a scaling coefficient of the expected correlations between related species on the tree, and varies from 0 (no correlation due to absence of phylogenetic signal) to 1 (covariance proportional to phylogenetic distance, implying maximal phylogenetic signal). Blomberg's *K* and its multivariate generalization $K_{mult}$ inform on how precisely the variance-covariance patterns found in the data are matched by the phylogenetic tree and where variance accumulates: $K \approx$ one implies that the mode of evolution closely resembles that expected under Brownian motion; when $K < 1$, close relatives resemble each other less than expected (variance is accumulated within clades), implying an evolutionary pattern that departs from a purely stochastic model (which could be caused by adaptation uncorrelated with phylogeny, that is homoplasy); finally, $K > 1$ is found when close relatives are more similar than expected under Brownian motion (variance is among clades), which could indicate stabilizing selection.

### Phylomorphospace and ancestral states analysis

To quantify major patterns of vestibular shape variation along the branches of the phylogeny we relied on a phylomorphospace approach (*Sidlauskas, 2008*), which allows us to intuitively visualize the extent and direction of the inferred shape change by means of branch length and orientation. This method projects the phylogenetic tree (*Figure 13*) onto the tangent space computed from the bgPCA and estimates the position in the morphospace of the internal nodes (ancestral morphologies) via a maximum likelihood (ML) method for continuous characters (*Felsenstein, 1988*; *Schluter et al., 1997*) using the 'fastAnc' function of *phytools* v0.6–60 R package (*Revell, 2012*). Subsequently, the bgPC scores of the ancestral states are rotated and translated from the shape data back into the configuration space for interpolation and 3D visualization using Deformetrica 3 software.

## Acknowledgements

Access to the HPC resources of CALMIP supercomputing center were granted under the allocation 2016-[P1440] attributed to the AMIS Laboratory, Toulouse (France). This work has been funded by the Agencia Estatal de Investigación and European Regional Development Fund of the European Union (projects CGL2016-76431-P and CGL2017-82654-P, AEI/FEDER, EU), by the Ministerio de Economía y Empresa (grant BES-2015–071318 to AU), the Generalitat de Catalunya (CERCA Programme, and consolidated research groups 2017 SGR 86 and 2017 SGR 116 GRC), and the French CNRS. We deeply thank Sergio Almécija for helpful advice and discussion during the elaboration of this paper and José Braga for valuable suggestions. We are grateful to Lorenzo Rook for giving access to the *Oreopithecus bambolii* petrosal bone originally published in *Rook et al. (2004)*. Lynn Copes, Lynn Lucas, and the MCZ provided access to a part of the scans used in the study, originally appearing in *Copes et al. (2016)*, funding for the collection of which was provided by NSF DDIG #0925793, and a Wenner-Gren Foundation Dissertation Grant #8102 (both to Lynn Copes). The Evolutionary Studies Institute provided access to the *Australopithecus* sp. data originally appearing in *Beaudet (2019a)*, the collection of which was funded by PAST, DST-NRF, CoE-Pal, Claude Leon Foundation, IFAS, French embassy, Leakey Foundation, Wits University. The scans for fossil and extant species were downloaded from MorphoSource.org, a web-accessible archive for 3D digital data housed by Duke University.

## Additional information

### Funding

| Funder | Grant reference number | Author |
| --- | --- | --- |
| Agencia Estatal de Investigación | CGL2016-76431-P | David M Alba |

| Agencia Estatal de Investigación | CGL2017-82654-P | Alessandro Urciuoli Salvador Moyà-Solà |
| Generalitat de Catalunya | 2017 SGR 86 | Alessandro Urciuoli Salvador Moyà-Solà |
| Generalitat de Catalunya | 2017 SGR 116 | David M Alba |
| Agencia Estatal de Investigación | BES-2015–071318 | Alessandro Urciuoli |

The funders had no role in study design, data collection and interpretation, or the decision to submit the work for publication.

### Author contributions
Alessandro Urciuoli, Conceptualization, Data curation, Software, Formal analysis, Investigation, Visualization, Methodology, Writing - original draft, Writing - review and editing; Clément Zanolli, Conceptualization, Data curation, Software, Methodology, Writing - review and editing; Amélie Beaudet, Data curation, Software, Formal analysis, Methodology, Writing - review and editing; Jean Dumoncel, Software, Methodology; Frédéric Santos, Data curation, Software, Methodology; Salvador Moyà-Solà, Formal analysis, Supervision, Funding acquisition, Writing - review and editing; David M Alba, Conceptualization, Data curation, Formal analysis, Supervision, Investigation, Writing - original draft, Writing - review and editing

### Author ORCIDs
Alessandro Urciuoli https://orcid.org/0000-0002-6265-8962
Clément Zanolli https://orcid.org/0000-0002-5617-1613
Amélie Beaudet https://orcid.org/0000-0002-9363-5966
Frédéric Santos https://orcid.org/0000-0003-1445-3871
David M Alba https://orcid.org/0000-0002-8886-5580

### Decision letter and Author response
Decision letter https://doi.org/10.7554/eLife.51261.sa1
Author response https://doi.org/10.7554/eLife.51261.sa2

## Additional files
### Supplementary files
• Supplementary file 1. List of extant anthropoid specimens analyzed in this paper. µCT image stacks were downloaded from MorphoSource digital repository at https://www.MorphoSource.org (Duke University, Durham, NC, USA) or scanned at Paul Sabatier University (PSU, Toulouse, France), the American Museum of Natural History (AMNH; New York, NY, USA) and Centro Nacional de Investigación sobre la Evolución Humana (CENIEH; Burgos, Spain).

• Transparent reporting form

### Data availability
All the results generated during this study are included as in the manuscript and as supporting files. Source data files are provided for Figures 2, 7, 8, and 9.

The following previously published datasets were used:

| Author(s) | Year | Dataset title | Dataset URL | Database and Identifier |
| --- | --- | --- | --- | --- |
| Beaudet A, Clarke RJ, Bruxelles L, Carlson KJ, Crompton R, de Beer F, Dhaene J, Heaton JL, Jakata K, Jashashvili T, Kuman K, McCly- | 2019 | Sterkfontein | https://www.morpho-source.org/Detail/Pro-jectDetail/Show/project_id/632 | MorphoSource, 632 |

| | | | | |
|---|---|---|---|---|
| mont J, Pickering TR, Stratford D | | | | |
| Copes LE, Lucas LM, Thostenson JO, Hoekstra HE, Boyer DM | 2016 | A collection of non-human primate computed tomography scans housed in MorphoSource, a repository for 3D data | https://www.morpho-source.org/Detail/Pro-jectDetail/Show/project_id/116 | MorphoSource, 116 |

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
