## [Decision Letter]

**Acceptance summary:**

This paper provides a comprehensive three-dimensional analysis of the inner-ear morphology of extant primates, with a focus on understanding the evolutionary history of apes and humans. As one of the hardest parts of the skull, the inner is often well-preserved in fossils and associated with diagnostic parts of the anatomy such as teeth. This makes it a useful candidate for the reconstruction of evolutionary patterns from fossil evidence, but first it is necessary to establish the degree to which it preserves a signal of these patterns. This paper provides such evidence with its analysis of living primates, and then draws in additional evidence from three fossil specimens: two putative human ancestors (*Australopithecus*) and one much older ape (*Oreopithecus*) with an unresolved evolutionary history. In addition to describing how inner-ear morphology varies along the branches of ape evolution, this paper will be of interest to researchers who are interested in the morphometric technique that was used to execute the analysis. The authors were able to describe shape variation in a more objective manner than more traditional use of set anatomical landmarks and semilandmarks, and this has potential application across any field of study that relies on the quantification and comparison of morphology.

**Decision letter after peer review:**

Thank you for sending your article entitled "The evolution of the vestibular apparatus in apes and humans" for peer review at *eLife*. Your article is being evaluated by three peer reviewers, one of whom is a member of our Board of Reviewing Editors, and the evaluation is being overseen by Diethard Tautz as the Senior Editor.

Summary:

This is a dense, technical manuscript that addresses the topic of morphological variation of the vestibular apparatus, and advances knowledge of key aspects of the evolution of anthropoid primates, especially hominoids. It is successful in demonstrating the utility of the bony labyrinth in preserving a phylogenetic signal across a relatively large sample of diverse, but also closely related, taxa. Both the sample and the methodology used are adequate and novel, and allow the authors to rigorously address the problem in question. Therefore, the results and conclusions reached (the establishment of the validity of the use of the vestibular apparatus, the proposal of characters and their phylogenetic polarity, and the description of the morphology of the LCAs for the different groups considered) are solidly justified and will serve other authors in future works.

However, this is also a highly specialized issue in primate evolution. Its broader significance to a larger readership remains unclear, even though the manuscript hints at larger implications for resolving fossil hominoid phylogeny, which is both controversial and has bearing on understanding the origins of the hominin clade. Although it may seem hominin-centric to place the significance of this manuscript with our own origins, that is certainly where its broader appeal will lie. The lack of such data, indeed the lack of more explicit emphasis on this issue in the discussion, makes this manuscript appear to be better suited in its present form for a more specialized journal. Although the reviewers appreciated the authors' caution in not overstating the significance of their results, a common concern was the broader appeal of the work as it presently stands.

Because of this ambiguity, in order to be published in *eLife*, the authors should do the following:

1) Provide a fossil test of the method they present here, preferably with Miocene hominoids, so that its broader applicability is very clear. The Abstract hints that some of this work may already be underway; however, if that is not the case and the authors do not already have those data and analyses in hand, then they should submit this version of the manuscript to a more specialized journal and consider *eLife* as a venue for their fossil application. This is because a typical revision turnaround time should be no more than about two months.

2) The authors should make their data presentation clearer, so that the differences between groups are explained in a very straightforward way. They should also amend their figures so that they are more easily readable, and correct some errors in the tables. These points are summarized below:

a) Methods: The authors should justify and explain their sample selection more in that part of the methods. For example, what is the validity for exploring the platyrrhines as a whole compared to, say the hominids as a whole (for example in Table 3), when the groups are not at the same taxonomic level? Also, throughout the text, and in the Materials and methods section, the definitions and measurement method of the SC-volume or SC-length variables appear to be missing. Since these variables are used in an important way in the manuscript, both variables must be adequately defined and also explained in detail how they have been measured so that other authors can measure them in the same way. A timeline for the proposed split times between the different analytical units that went into the analysis (based on both genetic and fossil estimates would be ideal) would be most helpful.

b) Fossil application: As for application to future fossil data, there are issues applying these results to the fossil record (as the authors note). It is not clear that stem hominines, for example, will follow the same pattern as crown hominines. If they do not, is this evidence that they are not in fact hominines, or simply that the synapomorphy of crown hominines had not yet evolved in early hominines? There is also some evidence of overlap in all the variables described. How is this quantified? What are the ranges of variation? To assess the phylogenetic signal in a fossil this variability needs to be quantified. Would it not be possible to do a DFA in order to assess the degree to which membership in a particular group is robust and predictable? What percentage of the members of each group are correctly identified? Certainly I would expect such an analysis if fossils are added to the analysis.

c) Figures require adjustment:

i) Currently, Figure 3—figure supplement 1 is the same as Figure 3. According to the text, Figure 3—figure supplement 1 should include hominoids and other anthropoids in superior view, but only includes hominoids in lateral view.

ii) Include in Figure 4 the values of R^2^ and the corresponding p values. It can make reading easier.

iii) A key directly on some figures would be extremely helpful (especially for Figure 6). This would make for a more understandable (visual) summary of the major differences between groups. At the end of the caption of Figure 6 it is said that humans can be distinguished by their more positive values but it seems that the distribution of the only four individuals of *Homo* is very similar to that of *P. troglodytes* and, perhaps, it would be better to remove this comment on humans from the caption of Figure 6.

iv) It would be easier to understand Figure 8 if the main morphological characteristics of each LCAs were indicated by arrows (or otherwise) in each of them.

d) Tables require adjustment:

i) In Table 2 caption it should say "as well as hominids and non-hominids separately" instead of "as well as hominoids and non-hominoids separately", since Table 2 only includes the terms hominids and non-hominids.

ii) So that the characters in Table 5 can be properly used by other authors, it would be very useful if the character states were illustrated with images.

iii) In Table 6 there is an error in the number of orangutan males: it should say 2 not 0, judging by the total of individuals (6) and the number of females (4).

iv) The narrative descriptions of the overall shape differences are clearly-written, but it would be easier to conceptualize and follow if there was less text and a single figure or set of figures with descriptive arrows highlighting the major differences. Such a figure may also include proposed or possible functional reasons for the differences, so that the visual form can be immediately linked to its potential adaptive purpose (where one may plausibly exist). A table summarizing some of these differences would also be a useful way to consolidate some of these results into a more digestible (and, frankly, useable) presentation.

---

## [Author Response]

Because of this ambiguity, in order to be published in eLife, the authors should do the following:1) Provide a fossil test of the method they present here, preferably with Miocene hominoids, so that its broader applicability is very clear. The Abstract hints that some of this work may already be underway; however, if that is not the case and the authors do not already have those data and analyses in hand, then they should submit this version of the manuscript to a more specialized journal and consider eLife as a venue for their fossil application. This is because a typical revision turnaround time should be no more than about two months.

As suspected by the editors, we are currently working on this approach with the aim to apply to extinct primate taxa. However, these fossils cannot be included in the present manuscript because they are unpublished and their analysis must be accompanied by detailed descriptions and comparisons outside the scope of this paper. However, to address the aforementioned request we have included in our analyses three specimens from two extinct hominoid species whose inner ear morphology is available from the literature. The addition of these taxa, which include a putative stem hominoid of uncertain affinities (*Oreopithecus*) and two specimens of early hominins (*Australopithecus),* support the reliability and usefulness of our approach. On the one hand, *Oreopithecus bambolii* (BAC 208) is an enigmatic late Miocene ape from Italy, whose affinities have been largely discussed during the last decades. Although there is currently a consensus that it is a hominoid, in the past it was even argued to be a cercopithecoid, and it is currently uncertain whether it is more closely related to early Miocene stem hominoids from Africa or to late Miocene stem hominids (dryopithecines) from Europe. Our analyses allow us to test among these competing hypotheses, thereby illustrating the potential of our approach for clarifying the phylogenetic affinities of Miocene hominoids. On the other hand, the vestibular apparatus morphology of two recently published individuals of South African *Australopithecus* (StW 578 and StW 573) enable testing the reliability of our approach, because the phylogenetic position of *Australopithecus* is well known (more closely related to humans than to other extant hominoids) and, hence, it can be predicted a priori that *Australopithecus* will show some derived features of *Homo* (as previously noted by the landmark-based approach published by Beaudet et al., 2019). Note that we included Amélie Beaudet as a coauthor due to her contribution to the analysis and interpretation of the *Australopithecus* vestibular morphology. Regarding *Oreopithecus*, our results support the hypothesis that it is a late-surviving stem hominoid and recover greatest similarities between australopiths and humans. Overall, we therefore think that the addition of these fossils greatly strengthens our conclusions and that the revised manuscript is more appealing to a broader readership than the original version.

2) The authors should undertake the recommendations of the reviewers to make their data presentation clearer, so that the differences between groups are explained in a very straightforward way. They should also amend their figures so that they are more easily readable, and correct some errors in the tables. These points are summarized below:

This has been done following the reviewers’ recommendations; see below for further details.

a) Methods: The authors should justify and explain their sample selection more in that part of the methods. For example, what is the validity for exploring the platyrrhines as a whole compared to, say the hominids as a whole (for example in Table 3), when the groups are not at the same taxonomic level?

Sample composition and selection have been justified in greater detail in the methods section of the revised version. The platyrrhine taxa included in this study only serve as an outgroup for catarrhines, i.e., to evaluate the polarity of evolutionary change. The description and analysis of vestibular morphology variability among New World monkeys is beyond the scope of this analysis, which is focused on hominoids. A different sample selection, more extensively sampling platyrrhine species and individuals, would be required to analyze the evolution of vestibular morphology among various platyrrhine subclades.

Also, throughout the text, and in the Materials and methods section, the definitions and measurement method of the SC-volume or SC-length variables appear to be missing. Since these variables are used in an important way in the manuscript, both variables must be adequately defined and also explained in detail how they have been measured so that other authors can measure them in the same way.

We concur with the reviewer’s request. Therefore, in the revised version we have amended the Materials and methods section by adding a definition for the SC volume and length variables. Based on these definitions, other researches should be able to take the same measurements and replicate our analyses.

A timeline for the proposed split times between the different analytical units that went into the analysis (based on both genetic and fossil estimates would be ideal) would be most helpful.

In the original version of the manuscript we already provided such information below the evolutionary tree depicted in Figure 9 (currently Figure 13), to which we have now added the two extinct hominoid genera incorporated into the revised version. To be more clear-cut in this regard, in the Materials and methods section of the revised version we have explicitly noted that such information is provided in the figure, and the uncertainty ranges of such datings are provided in several tables published by Springer et al., 2012. In case the reviewer considers it necessary, we could provide a table with the mean divergence rates and 95% composite credibility intervals, but we think it is not necessary, as the information is available from the literature.

b) Fossil application: As for application to future fossil data, there are issues applying these results to the fossil record (as the authors note). It is not clear that stem hominines, for example, will follow the same pattern as crown hominines. If they do not, is this evidence that they are not in fact hominines, or simply that the synapomorphy of crown hominines had not yet evolved in early hominines?

The main problem with fossil specimens is that their phylogenetic relationships are often controversial, so that different phylogenetic placements should be inspected to assess if the implications of the fossil taxa to the reconstruction of ancestral conditions varies. This problem, however, does not apply to extinct taxa of well-known phylogenetic relationships (such as *Australopithecus*, see below). Furthermore, the synapomorphies recognized based on the living taxa should contribute to further clarify the position of more controversial extinct taxa (such as *Oreopithecus*, see also below), based on the possession or absence of shared-derives features of extant clades. Of course, extinct taxa might eventually show that some of these synapomorphies are indeed homoplasies independently evolved among various extant hominoid lineages, or that some independently evolved on some extinct taxa. However, the condition displayed by the extant forms should constitute the baseline to assess the position of the extinct forms.

As for the specific concern by the reviewer about hominines, a priori, a stem hominine would be expected to display some (but not necessarily all) of the synapomorphies of the hominine crown group, while at the same time lacking apomorphies of any hominine subclade. In contrast, a crown hominine would be expected to display all of the synapomorphies of the hominine crown group plus some derived features of one of the hominine subclades. The same applies for other groups. In the case of our analysis of inner ear morphology, the ancestral condition reconstructed for crown hominines closely resemble that for crown hominids as a whole, making it more difficult a priori to discern hominids from stem hominines than from pongines on this basis alone. But if a particular extant taxon shows some but not all synapomorphies of a particular crown group, then it must be assumed it is a member of the stem lineage (unless further analyses show some of the purported synapomorphies to have independently evolved among extant members of the group). In the revised version, we have been more explicit about the presence/absence of crown synapomorphies when interpreting extinct taxa, and have exemplified this when discussing the two extinct genera added to the revised version.

There is also some evidence of overlap in all the variables described. How is this quantified? What are the ranges of variation? To assess the phylogenetic signal in a fossil this variability needs to be quantified. Would it not be possible to do a DFA in order to assess the degree to which membership in a particular group is robust and predictable? What percentage of the members of each group are correctly identified? Certainly I would expect such an analysis if fossils are added to the analysis.

Unlike classical morphometric methods based on linear measurements or shape ratios, it makes no sense to quantify variation in the original variables as they represent mathematical moments, that could be assimilated to a kind of "data coordinates". It is only thanks to composite multivariate analyses such as PCA and bgPCA that these data can be interpreted. In this regard, the overlap among groups can be assessed (even if not quantified) from the bivariate plots of the various PCs explaining most of the variance. We concur with the reviewer that it would be useful to assess the discrimination among groups by computing the number of original correctly classified specimens. Because of this reason, the reviewer suggests to perform a discriminant analysis, which would further enable to compute the closest similarities of the fossil specimens as compared to the extant taxa. The problem is that discriminant analyses are designed to maximize the distinction among the a priori defined groups and, to do so, they distort the morphospace in a way that tends to exaggerate a lot the group distinction. Furthermore, for mathematical reasons related to the way a discriminant analysis is computed, it is required that the number of variables is lower than the number of specimens. The usual way to circumvent this issue is to use a limited number of PC scores as variables. Nevertheless, the choice of the most adequate number of PC scores is questionable. For this reason, we rely instead on a classification based on the between-group PCA, which is similarly based on intergroup variance but does not distort so much the morphospace to maximize classification accuracy. The classification results with cross-validation have been added to the new Table 1 and reported for fossils in the new Tables 3 and 4; the Results, Material and Methods, and Discussion sections have been amended accordingly. The cross-validated classification and the posterior probabilities results were computed by Frédéric Santos, which has been accordingly added as a coauthor.

c) Figures require adjustment:i) Currently, Figure 3—figure supplement 1 is the same as Figure 3. According to the text, Figure 3—figure supplement 1 should include hominoids and other anthropoids in superior view, but only includes hominoids in lateral view.

We thank the reviewers for bringing this issue to our attention. It was our inadvertent mistake during the submission process. We have changed the file for Figure 3—figure supplement 1 with the correct one. Figure 3—figure supplement 1 now depicts the vestibular superior view in both extant and extinct hominoids. In addition, since *eLife* is not restrictive in the use of figures, but rather encourages it for enhancing the comprehensibility of the text, we have moved Figure 3—figure supplements 1 and 2 to Figure 4 and Figure 5, respectively.

ii) Include in Figure 4 the values of R^2^ and the corresponding p values. It can make reading easier.

We have added the R^2^ and *p* values for each regression to Figure 4 (now Figure 7). The text has been colored to match that of the corresponding regression line.

iii) A key directly on some figures would be extremely helpful (especially for Figure 6). This would make for a more understandable (visual) summary of the major differences between groups.

We thank the reviewer for bringing this issue to our attention. Accordingly, we have implemented the key legend for Figure 5 (now Figure 8) and added convex hulls to Figures 2 and 6 (the latter now Figure 9), in order to help readers to identify the distribution of the groups distinguished in the bgPCA plots, as well as to identify the species to which each symbol corresponds.

At the end of the caption of Figure 6 it is said that humans can be distinguished by their more positive values but it seems that the distribution of the only four individuals of Homo is very similar to that of P. troglodytes and, perhaps, it would be better to remove this comment on humans from the caption of Figure 6.

Done.

iv) It would be easier to understand Figure 8 if the main morphological characteristics of each LCAs were indicated by arrows (or otherwise) in each of them.

Following the reviewer’s suggestion, we modified Figure 8 (now Figure 11). In particular, we implemented a pseudocolor map for the 3D meshes, which visually explains the zones of maximum and minimum deformation for the surface. Furthermore, we added a set of vectors (indicated by arrows of different size) that correspond to the magnitude and direction of the deformation. We think that these additions will be very helpful for readers to identify and visualize the characters that we describe within the main text.

d) Tables require adjustment:i) In Table 2 caption it should say "as well as hominids and non-hominids separately" instead of "as well as hominoids and non-hominoids separately", since Table 2 only includes the terms hominids and non-hominids.

Corrected (note that former Table 2 is now Table 5).

ii) So that the characters in Table 5 can be properly used by other authors, it would be very useful if the character states were illustrated with images.

Following the reviewer’s suggestion, we added a new figure (Figure 12) that summarizes and visually illustrates the proposed synapomorphies to the readers, based on the information reported in former Table 5 (now Table 8).

iii) In Table 6 there is an error in the number of orangutan males: it should say 2 not 0, judging by the total of individuals (6) and the number of females (4).

We thank the reviewer for bringing this issue to our attention. We amended Table 6 (now Table 9) by adding a column for individuals of unknown sex. This is because two of the *Pongo* specimens included in this work are represented by CT scans of juvenile individuals’ petrosals, thus preventing us from determining their sex.

iv) The narrative descriptions of the overall shape differences are clearly-written, but it would be easier to conceptualize and follow if there was less text and a single figure or set of figures with descriptive arrows highlighting the major differences. Such a figure may also include proposed or possible functional reasons for the differences, so that the visual form can be immediately linked to its potential adaptive purpose (where one may plausibly exist).

We appreciate the reviewer’s suggestion to improve the clarity of the morphological descriptions by means of images. Accordingly, as stated above (see point no. 2), we added a new figure that summarized the most important morphological differences among the analyzed taxa (see Figure 12). Furthermore, major differences are highlighted by the deformation maps newly reported in Figures 3, 4, 5 and 6, which illustrate the magnitude and direction of the changes by means of vectors (depicted by arrows). In contrast, we refrained from adding potential functional links in any of the figures, not only because it would complicate their readability, but also to present an oversimplified functional interpretation of the features discussed. We consider that caution must be exercised when inferring locomotor repertoires from vestibular morphology, so we prefer to refer the readers to the more detailed explanations provided in the main text. Furthermore, the main aim of our manuscript is rather to highlight the phylogenetic significance of the various features of the vestibular apparatus, irrespective of whether their functional meaning is more or less well known.

A table summarizing some of these differences would also be a useful way to consolidate some of these results into a more digestible (and, frankly, useable) presentation.

We consider that our former Table 5 (currently Table 8) already summarizes the main differences between hominoids and non-hominoids, and also between hylobatids and hominids, in a useful way. The latter is particularly true in the sense that characters are defined and scored in a cladistic manner, which greatly facilitates their inclusion in future cladistic analyses as well as their applicability for discussion the phylogenetic placement of the extinct taxa analyzed. The only information that is missing from this table are autapomorphic features of extant hominoid taxa, but we do not consider it necessary to summarize such information in a similar manner – particularly because autapomorphic features are not phylogenetically informative from the viewpoint of parsimony analysis. Therefore, we did not implement further changes to address this particular suggestion.